# EZH2 controls epicardial cell migration during heart development

Haobin Jiang[1,3,*], Lina Bai[1,*] ⓘ, Shen Song[1] ⓘ, Qianqian Yin[1], Anteng Shi[1], Bin Zhou[4], Hong Lian[1], Houzao Chen[5], Cheng-ran Xu[6] ⓘ, Yanchun Wang[7], Yu Nie[1,2] ⓘ, Shengshou Hu[1]

Enhancer of zeste homolog 2 (EZH2) is an important transcriptional regulator in development that catalyzes H3K27me3. The role of EZH2 in epicardial development is still unknown. In this study, we show that EZH2 is expressed in epicardial cells during both human and mouse heart development. *Ezh2* epicardial deletion resulted in impaired epicardial cell migration, myocardial hypoplasia, and defective coronary plexus development, leading to embryonic lethality. By using RNA sequencing, we identified that EZH2 controls the transcription of tissue inhibitor of metalloproteinase 3 (TIMP3) in epicardial cells during heart development. Loss-of-function studies revealed that EZH2 promotes epicardial cell migration by suppressing TIMP3 expression. We also found that epicardial *Ezh2* deficiency–induced TIMP3 upregulation leads to extracellular matrix reconstruction in the embryonic myocardium by mass spectrometry. In conclusion, our results demonstrate that EZH2 is required for epicardial cell migration because it blocks *Timp3* transcription, which is vital for heart development. Our study provides new insight into the function of EZH2 in cell migration and epicardial development.

## Introduction

The epicardium, a mesothelial layer covering all vertebrate hearts, plays a critical role in cardiac development by acting as a hub of multiple cardiac cell lineages (Quijada et al, 2020). During heart development, epicardial cells migrate into the myocardium to form epicardial-derived cells (EPDCs), which give rise to various cardiac lineages, including fibroblasts, vascular smooth muscle cells, pericytes, and endothelial cells, with controversial contributions to cardiomyocytes (Zhou et al, 2008; Cao et al, 2020). Loss of EPDCs in the myocardium leads to impaired cardiac function due to myocardial hypoplasia and abnormal vascular plexus formation (von Gise et al, 2011; Diman et al, 2014; Singh et al, 2016). The migration of epicardial cells from the outermost layer into the myocardium is an important prerequisite for successful cell differentiation of EPDC (Singh et al, 2016; Liu et al, 2018). However, the mechanisms and functional mediators of epicardial cell migration need further exploration.

Enhancer of zeste homolog 2 (EZH2), the major histone methyltransferase of polycomb repressor complex 2 (PRC2), establishes the trimethylation of histone H3 at lysine 27 (H3K27me3), an epigenetic mark that functions to maintain transcriptional repression (Rosa-Garrido et al, 2018). Deletion of *Ezh2* in cardiac progenitor cells results in congenital heart malformations due to cardiomyocyte proliferation inhibition (He et al, 2012). In addition, inactivation of EZH2 in cardiac progenitors of the anterior heart field causes postnatal myocardial pathology attributed to abnormal cardiomyocyte differentiation (Delgado-Olguín et al, 2012). Myocardial inactivation of EZH2 blocks cardiomyocyte proliferation and abolishes the regenerative ability of neonatal hearts (Yue et al, 2019). Single-cell RNA sequencing has revealed that EZH2 is detectable in human embryonic epicardial cells (Cui et al, 2019), indicating that EZH2 might be involved in the development of epicardium during cardiac development.

In this study, we found that EZH2 was expressed in both human and mouse epicardial cells during heart development. Epicardial deletion of *Ezh2* led to embryonic lethality and epicardial cell migration inhibition. By using RNA sequencing (RNA-Seq) and ECM mass spectrometry, we identified that EZH2 influenced epicardial cell migration into the myocardium via TIMP3-dependent fine tuning of basement membrane degradation, which is vital for heart development. Our results demonstrate a critical role of EZH2 in epicardium and heart development.

[1]State Key Laboratory of Cardiovascular Disease, Fuwai Hospital, National Center for Cardiovascular Disease, Chinese Academy of Medical Sciences and Peking Union Medical College, Beijing, China   [2]National Health Commission Key Laboratory of Cardiovascular Regenerative Medicine, Fuwai Central-China Hospital, Central-China Branch of National Center for Cardiovascular Diseases, Zhengzhou, China   [3]Department of Thoracic Surgery, The First Affiliated Hospital, Zhejiang University School of Medicine, Hangzhou, China   [4]State Key Laboratory of Cell Biology, CAS Center for Excellence in Molecular Cell Science, Institute of Biochemistry and Cell Biology, Shanghai Institutes for Biological Sciences, University of Chinese Academy of Sciences, Chinese Academy of Sciences, Shanghai, China   [5]State Key Laboratory of Medical Molecular Biology, Department of Biochemistry and Molecular Biology, Institute of Basic Medical Sciences, Chinese Academy of Medical Sciences & Peking Union Medical College, Beijing, China   [6]Department of Human Anatomy, Histology, and Embryology, School of Basic Medical Sciences; Peking-Tsinghua Center for Life Sciences, Peking University, Beijing, China   [7]Haidian Maternal & Child Health Hospital, Beijing, China

Correspondence: nieyuniverse@126.com; huss@fuwaihospital.org
*Haobin Jiang and Lina Bai contributed equally to this work

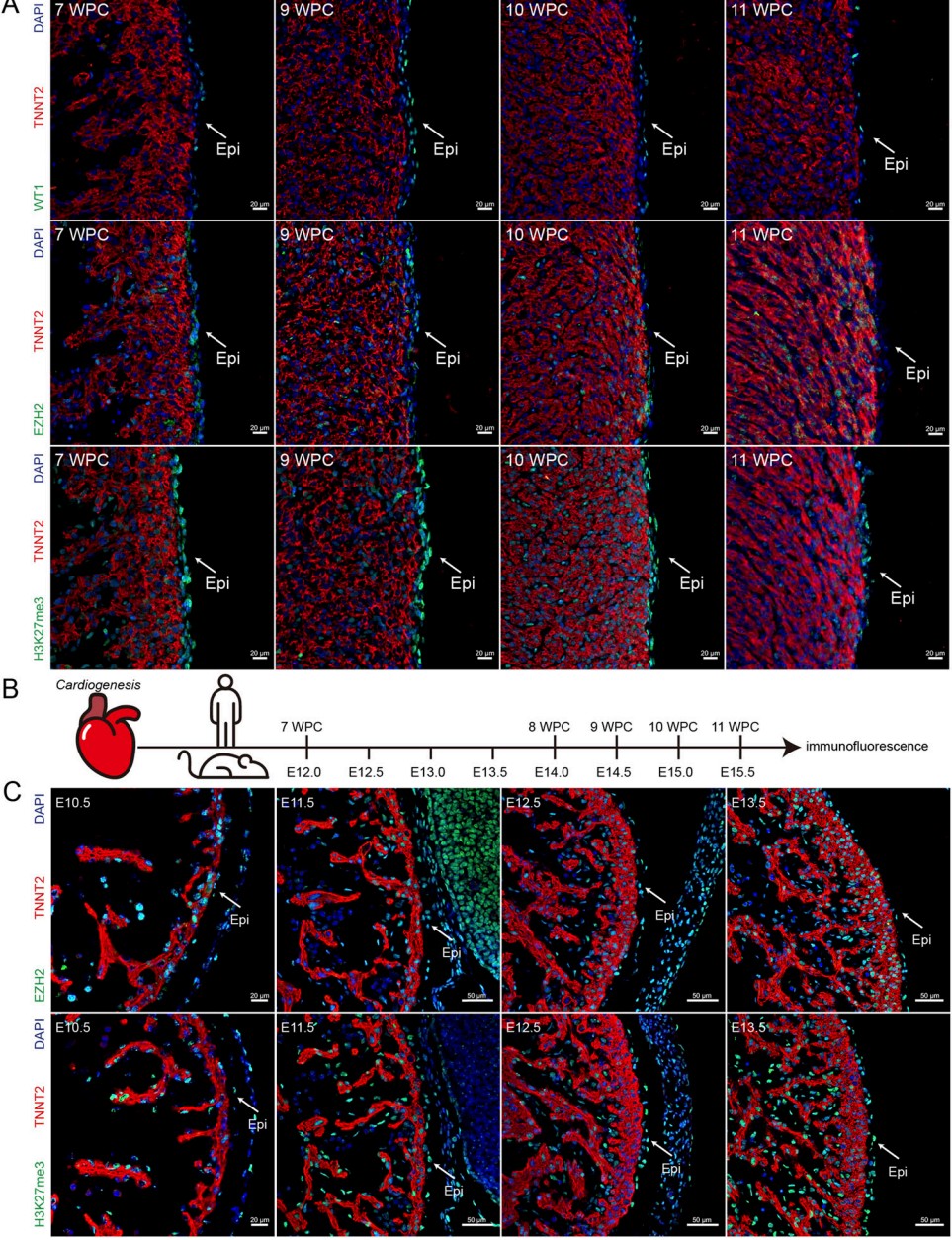

**Figure 1. EZH2 expression in human and mouse embryonic epicardium.**
**(A)** Immunostaining for WT1 (green), EZH2 (green), H3K27me3 (green), TNNT2 (red), and DAPI (blue) in human heart sections. Scale bar, 20 μm. **(B)** Development stage correspondences between humans and mice (weeks post conception, WPC). **(C)** Immunostaining for EZH2 (green), H3K27me3 (green), TNNT2 (red), and DAPI (blue) in mouse heart sections. Scale bar, left 20 μm; right 50 μm.

# Results

## EZH2 is involved in human and mouse epicardial development

EZH2 is a specific histone methyltransferase of histone H3 at Lys 27 (H3K27), and EZH2 catalyzes the trimethylation of histone 3 lysine 27 (H3K27me3) (Zovoilis et al, 2016), leading to the repression of gene transcription and regulating proliferation and differentiation in early embryonic development (Duan et al, 2020). To illustrate the pattern of EZH2 expression during human epicardial development, we performed EZH2 and H3K27me3 immunohistochemistry on human embryonic hearts at 7–11 wk post conception. We found that Wilms tumor 1 (WT1, a marker of epicardial cells), EZH2, and H3K27me3 were detectable in human epicardial cells during heart development (Fig 1A).

To confirm whether mice could be used as animal models to study the role of EZH2 in epicardial development, we collected mouse embryonic hearts and examined the expression of EZH2 and H3K27me3 in epicardial cells by immunostaining (Fig 1B). Our results showed that EZH2 and H3K27me3 were readily detectable in mouse epicardial cells from E10.5 to E13.5 (Fig 1C), which suggests that the expression of EZH2 in epicardial development is conserved in humans and mice.

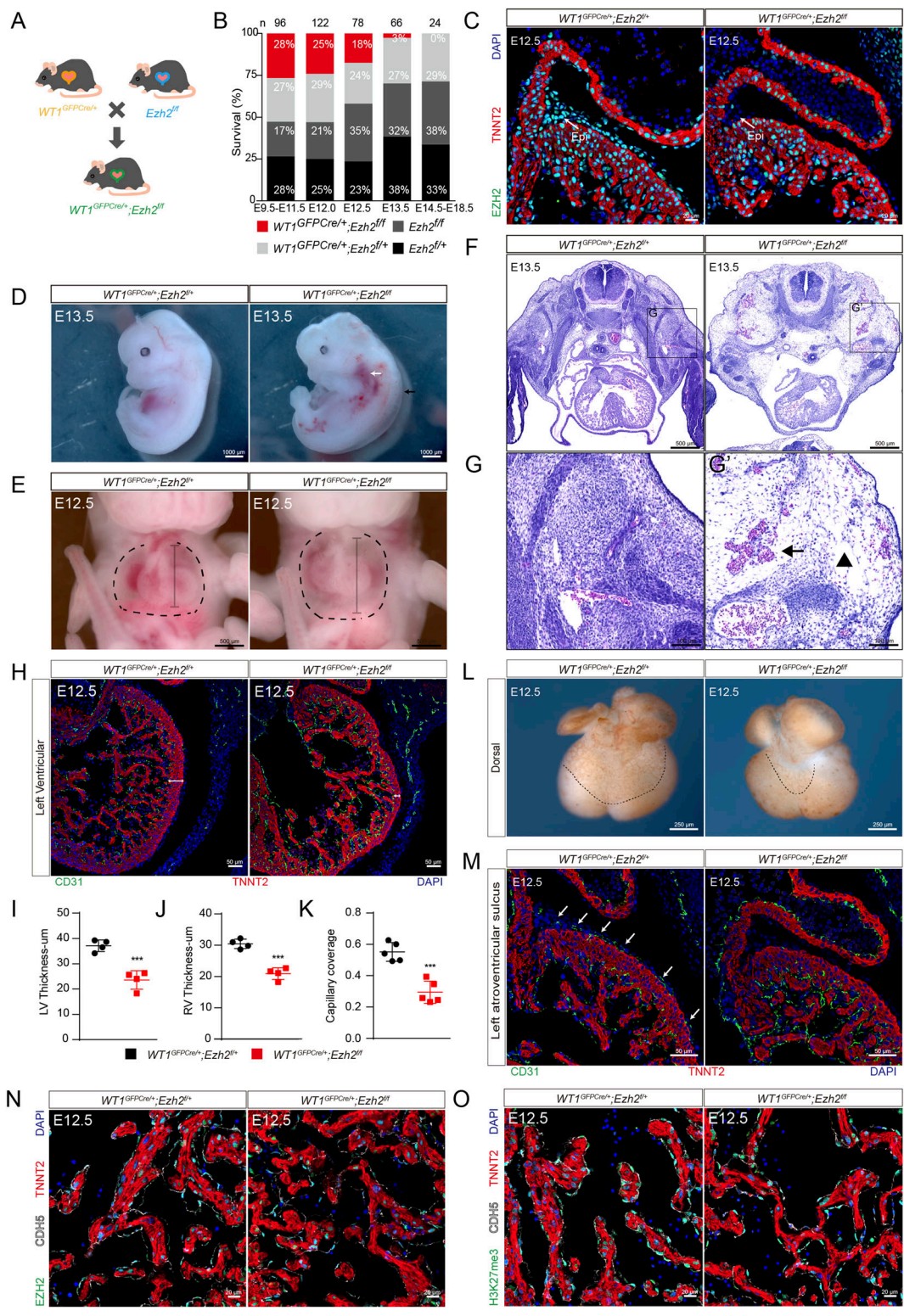

**Figure 2. Myocardial hypoplasia and defective coronary plexus development in *Ezh2^epi-KO* mice.**
**(A)** Schematic presentation describing the generation of epicardium conditional knockout of *Ezh2* mice. **(B)** The survival ratio of embryos with all genotypes was generated by crossing the *WT1^GFPCre/+*;*Ezh2^flox/+* and *Ezh2^flox/flox* mice line. Most *Ezh2^epi-KO* embryos died between E12.5–E13.5. Numbers above bars indicate sample size. **(C)** Immunostaining for EZH2 (green), TNNT2 (red), and DAPI (blue) in mouse heart sections. Scale bar, 20 μm. **(D)** *Ezh2^epi-KO* embryos displayed passive congestion (white arrow) and cutaneous edema (black arrows). Scale bar, 1,000 μm. **(E)** *Ezh2^epi-KO* embryos displayed pronounced pleural effusion. Scale bar, 500 μm. **(F)** H&E transverse sections showed that *Ezh2^epi-KO* hearts were smaller. Scale bar, 500 μm. **(G)** H&E transverse sections showed that *Ezh2^epi-KO* embryos displayed passive congestion (black arrow) and cutaneous edema (black arrowhead). Scale bar, 100 μm. **(H)** Immunostaining for CD31 (green), TNNT2 (red), and DAPI (blue) in mouse heart sections. Scale bar,

## Epidermal deletion of *Ezh2* leads to myocardial hypoplasia and defective coronary plexus development

To assess the potential role of EZH2 in the developing epicardium, we generated an epicardium specific loss-of-function mutant by crossing *Ezh2^flox/flox* mice with the *WT1^GFPCre/+* mice (Fig 2A). By E12.5, *WT1^GFPCre/+; Ezh2^flox/flox* (hereafter referred to as *Ezh2^epi-KO*) epicardial cells exhibited markedly reduced EZH2 immunoreactivity (Fig 2C). We did not recover any *Ezh2^epi-KO* neonates from the breeding of *WT1^GFPCre/+; Ezh2^flox/+* and *Ezh2^flox/flox* mice (Fig 2B), suggesting that epicardial in-activation of EZH2 is embryonically lethal. Further genotyping of embryos from timed mating showed that the loss of EZH2 resulted in embryonic lethality between E12.5 and E13.5 (Fig 2B).

At E12.5 and E13.5, *Ezh2^epi-KO* embryos showed remarkable hydrops fetalis, cutaneous edema, pleural effusion, and passive congestion (Fig 2D–G), consistent with embryonic lethality due to heart dysfunction. At E12.5, immunostaining for the cardiac marker TNNT2 and endothelial marker CD31 showed that ventricular wall thickness decreased significantly in *Ezh2^epi-KO* hearts (Fig 2H–J). In addition, *Ezh2^epi-KO* hearts exhibited markedly impaired coronary vascular development, as whole-heart CD31 staining showed a significant decrease in plexus density in E12.5 *Ezh2^epi-KO* hearts (Fig 2K and L), which was also verified by CD31 staining of tissue sections (Fig 2M). Meanwhile, we analyzed the endothelial cell–specific EZH2 knockout efficiency by co-immunostaining of EZH2/H3K27me3 with CDH5 in *Ezh2^epi-KO* mice. There was a large amount of EZH2⁺ or H3K27me3⁺ endothelial cells in *WT1^GFPCre/+;Ezh2^flox/+* and *Ezh2^epi-KO* hearts, and there was no significant difference between the two groups (Fig 2N and O). The results suggested that epicardial cell–specific EZH2 knockout had no effect on EZH2 expression in endothelial cells.

Furthermore, we constructed inducible EZH2 epicardial conditional knockout mouse by crossing *WT1^CreERT2/+* mouse line with *Ezh2^flox/flox*. Two doses of Tamoxifen were administered at E9.5 and E10.5, *WT1^CreERT2/+;Ezh2^flox/flox* (hereafter as *Ezh2^epi-iKO*) epicardial cells exhibited markedly reduced EZH2 immunoreactivity by E12.5 (Fig S1A and B). Immunostaining results showed a less density of intramyocardial vessels and decreased epicardial cell number (Fig S1C–F). This cardiac phenotype of *Ezh2^epi-iKO* mice is generally consistent with that of *Ezh2^epi-KO* mice.

Together, the results indicate that the phenotype of *Ezh2* deletion in WT1-Cre is only due to EZH2 loss in the epicardial lineage and not due to other cell types, such as endothelial cells, suggesting that EZH2 expression in the epicardium is indispensable during heart development.

## EZH2 is involved in epicardial cell migration

Epicardial cell migration into the myocardium to form EPDCs is a critical process for heart development (Zhou et al, 2008; Cao et al, 2020). Epicardial cell proliferation, apoptosis, and migration all affect the formation of EPDCs (Wu et al, 2010; Sun et al, 2021). In this study, EPDCs were undetectable in the compact myocardium in E12.5 *Ezh2^epi-KO* hearts, although EPDCs existed in *WT1^GFPCre/+; Ezh2^flox/+* hearts (Fig 3A); this finding indicates that EZH2 is involved in epicardial cell migration and EPDC formation. Terminal deoxy-nucleotidyl transferase (TdT) dUTP nick-end labeling (TUNEL) staining revealed that apoptosis was undetectable in both *Ezh2^epi-KO* and *WT1^GFPCre/+;Ezh2^flox/+* hearts, indicating that apoptosis is not involved in the decrease in EPDCs in the *Ezh2^epi-KO* myocardium (Fig 3B). We found that WT1-positive epicardial cell proliferation was significantly decreased in *Ezh2^epi-KO* hearts without apoptosis (Fig S2). In addition, *Ezh2^epi-iKO* epicardial cells exhibited markedly reduced cell migration by E12.5, which showed inhibited epicardial cell migration (Fig S1G). These data all indicate that EZH2 deficiency in vivo impedes epicardial cell migration.

To explore the role of EZH2 in regulating epicardial cell migration, we isolated primary epicardial cells from E11.5 mice and knocked down *Ezh2* with siRNA (Fig 3C and F). Immunofluorescence and qRT-PCR showed that we obtained high-purity epicardial cells through tissue attachment (Fig 3D and E). Then, we performed RNA sequencing (RNA-Seq) on *Ezh2*-deficient epicardial cells. Gene Ontology (GO) analysis showed that genes were enriched in terms of cell migration and development (Fig 3G). We performed explant heart culture and found a significantly decreased outgrowth of epicardial cells in *Ezh2^epi-KO* heart explants relative to *WT1^GFPCre/+; Ezh2^flox/+* hearts (Fig 3H and I). Scratch wound assays showed that the migration of primary epicardial cells was significantly inhibited in the *Ezh2* knockdown group (Fig 3J and K), indicating that EZH2 is involved in epicardial cell migration.

## TIMP3 is implicated in EZH2-induced epicardial cell migration

To unveil the mechanism of EZH2 in epicardial cell migration, we analyzed the gene expression profile in *Ezh2* knockdown primary epicardial cells based on RNA-Seq data. Among the up-regulated genes, tissue inhibitor of metalloproteinase 3 (Timp3) most significantly increased in *Ezh2* knockdown primary epicardial cells (–log10 *P*-value > 300) (Fig 4A). Consistent with the sequencing data, we observed marked elevation in *Timp3* expression in *Ezh2^epi-KO* epicardial explants by qRT-PCR and RNAscope (Fig 4B–D). The epicardial cells were cultured with an siRNA transfection kit to knock down *Ezh2* (Fig 4E). We found that *Ezh2* deletion could cause the increase of TIMP3 mRNA (Fig 4F) and protein level (Fig 4G). However, the expression of EZH2 was not affected by *Timp3* knockdown in epicardial cells (Fig 4H), suggesting that TIMP3 was a downstream effector of EZH2 during epicardium development.

The role of the H3K27 methyltransferase EZH2 in the methylation of H3K27 is well established, causing an increase in H3K27me3, and

50 *μm*. **(I, J)** Wall thickness of the left ventricular and right ventricular compact myocardium (n = 4). **(K)** Quantification of vascular coverage on the dorsal side of the ventricle in graph L (n = 5). **(L)** Whole-mount PECAM1 immunostaining. Scale bar, 250 *μm*. **(M)** *Ezh2^epi-KO* embryos displayed impaired vascular growth (white arrow). Scale bar, 50 *μm*. **(N)** Immunostaining for EZH2 (green), CDH5 (white), TNNT2 (red), and DAPI (blue) in mouse heart sections. Scale bar, 20 *μm*. **(O)** Immunostaining for H3K27me3 (green), CDH5 (white), TNNT2 (red), and DAPI (blue) on mouse heart sections. Scale bar, 20 *μm*. Data information: in (I, J, K), data are presented as mean ± SEM. NS, not significant; ∗*P* < 0.05, ∗∗*P* < 0.01, and ∗∗∗*P* < 0.001.

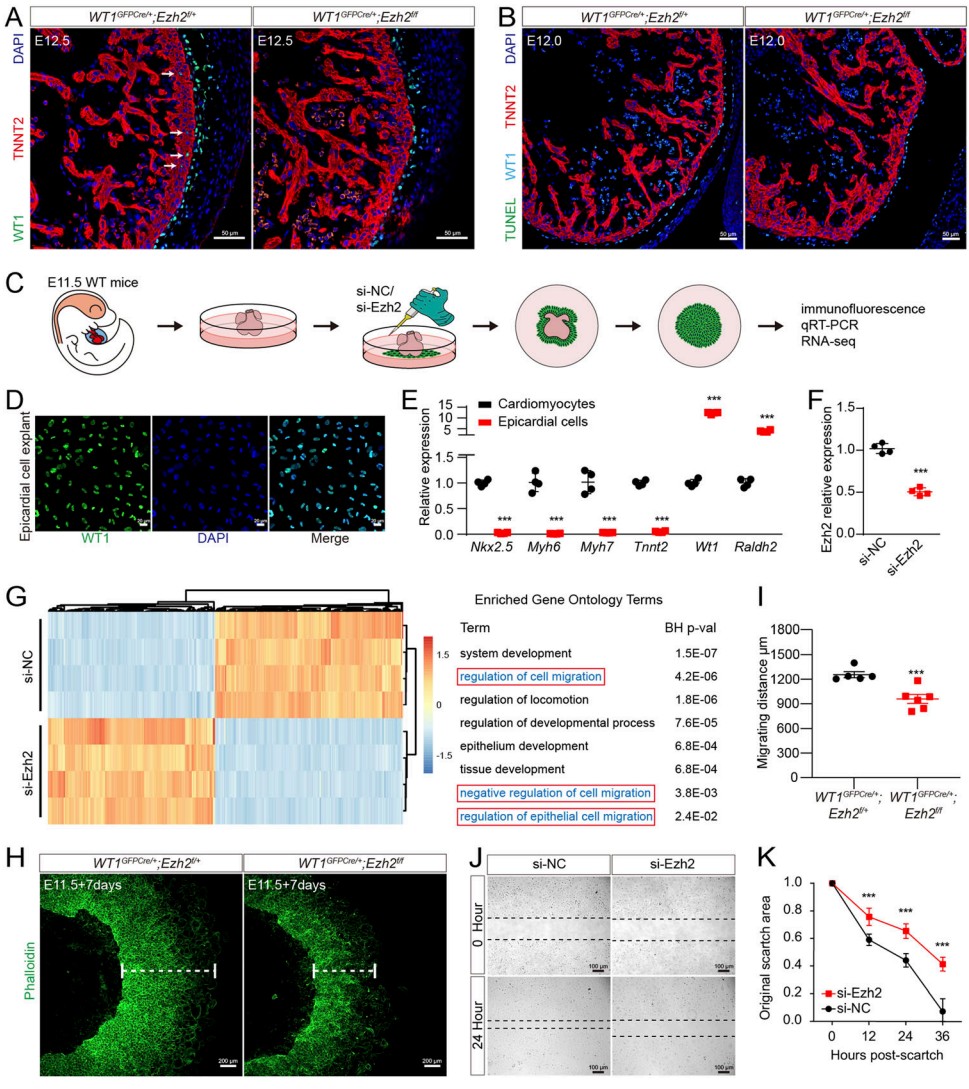

**Figure 3. EZH2 regulates epicardial cell migration.**
**(A)** Immunostaining for WT1 (green), TNNT2 (red), and DAPI (blue) in mouse heart sections. The white arrow indicates the EPDCs in the compact myocardium. Scale bar, 50 µm. **(B)** TUNEL staining of mouse heart sections. Scale bar, 50 µm. **(C)** Primary epicardial cells were isolated from E11.5 mice, and *Ezh2* was knocked down with siRNA. Immunofluorescence, qRT-PCR, and RNA-Seq were then performed. **(D)** The purity of primary epicardial cells was confirmed by immunostaining for WT1 (green). Scale bar, 20 µm. **(E)** qRT-PCR revealed robust expression of the epicardial marker genes *Wt1* and *Raldh2* and low expression of the cardiomyocyte genes *Nkx2-5*, *Myh6*, *Myh7*, and *Tnnt2* in primary epicardial cells (n = 4). **(F)** Interference efficiency 48 h after transfection (n = 4). **(G)** RNA-Seq identified differentially expressed genes and GO terms in the si-Ezh2 group and si-NC group. **(H)** Immunostaining for phalloidin (green) in mouse epicardial explants. Scale bar, 200 µm. **(I)** Migration in (H) is presented as cell migration width (n = 6 for each condition). **(J)** Scratch assay following knockdown of EZH2. Scale bar, 100 µm. **(K)** Migration in (J) is presented as a percentage of the original scratch area (n = 5 for each condition). Data information: in (E, F, I, K), data are presented as mean ± SEM. NS, not significant; *P < 0.05, **P < 0.01, and ***P < 0.001.

subsequently suppressing transcription of genes bound by such histones. To confirm how EZH2 regulates the expression of TIMP3, we next examined the presence of H3K27me3 to determine whether H3K27me3 might be involved in the process of epicardial cell migration. By E12.5, *Ezh2^epi-KO* epicardial cells exhibited markedly reduced H3K27me3 immunoreactivity (Fig 4I). EZH2 is the catalytic subunit of PRC2, whose methyltransferase activity requires two other subunits: embryonic ectoderm development and suppressor of zeste 12 (SUZ12). We next performed chromatin immunoprecipitation (ChIP) followed by quantitative PCR (ChIP–qRT-PCR) and found that H3K27me3 and Suz12 (PRC2 complex, which generally forms a core together with EZH2) were enriched at *Timp3* promoters of EpiSV40 (Fig 4J–L), an established cell line transformed from murine primary epicardial cells (Jiang et al, 2021). Together, we confirmed that *Timp3* expression is regulated by PRC2 complex–H3K27me3 modification, in which EZH2 plays a key role. Collectively, our results demonstrated that EZH2 suppresses TIMP3 expression by introducing H3K27me3 modification on the promoters.

To determine whether TIMP3 regulates epicardial cell migration, we treated primary epicardial cells with TIMP3 recombinant protein. Scratch wound assays showed that TIMP3 administration inhibited primary epicardial cell migration (Fig 5A and B). We also found that *Ezh2* deficiency–induced migration suppression was rescued by *Timp3* knockdown in epicardial cells (Fig 5C and D). Furthermore, we isolated primary epicardial cells from E12.5 *Ezh2^epi-KO* mice and knocked down *Timp3* with siRNA (Fig 5E). Immunofluorescence staining of phalloidin showed that *Timp3* knockdown reversed the epicardial migration suppression caused by *Ezh2* deletion (Fig 5F and G). These results indicate that EZH2 regulates epicardial cell migration via TIMP3.

**EZH2/TIMP3 regulates epicardial cell migration by degrading ECM**

TIMP3 is a strong inhibitor of a large range of matrix metalloproteases and disintegrin and metalloproteinase (ADAMs) family proteins, which are involved in degradation of the ECM (Jackson

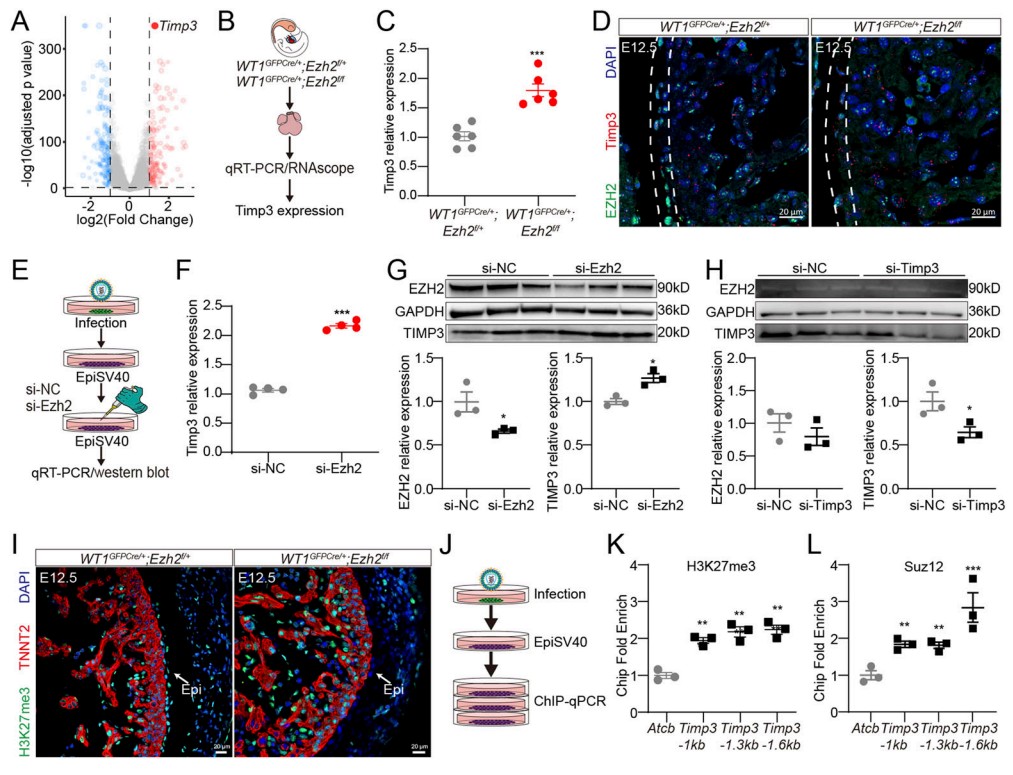

**Figure 4. EZH2 suppresses TIMP3 expression through H3K27me3 modification.**

**(A)** Volcano plots showing differentially expressed genes, *Timp3* most significantly increased in *Ezh2* knockdown primary epicardial cells. **(B)** *Timp3* expression was detected in *Ezh2^epi-KO^* epicardial explants by qRT-PCR and RNAscope. **(C)** Marked up-regulation of *Timp3* by qRT-PCR assay in *Ezh2^epi-KO^* epicardial explants (n = 6). **(D)** Marked up-regulation of *Timp3* by RNAscope assay in *Ezh2^epi-KO^* epicardial explants. **(E)** The epicardial cells (EpiSV40, an established cell line transformed from murine primary epicardial cells) were cultured with a siRNA transfection kit for another 3 d to knock down EZH2. **(F)** Up-regulation of *Timp3* mRNA level in si-Ezh2–treated epicardial cells (n = 4). **(G)** Immunoblotting for EZH2, TIMP3, and GAPDH in si-Ezh2–treated epicardial cells (n = 3). **(H)** Immunoblotting for EZH2, TIMP3, and GAPDH in si-Timp3–treated epicardial cells (n = 3). **(I)** Immunostaining for H3K27me3 (green), TNNT2 (red), and DAPI (blue) in mouse heart sections. Scale bar, 20 μm. **(J)** ChIP followed by quantitative PCR (ChIP–qRT-PCR) was performed in EpiSV40. **(K)** H3K27me3 ChIP–qRT-PCR for *Timp3* in EpiSV40 cells (n = 3). *Actb* was a negative control. **(L)** Suz12 ChIP–qRT-PCR for *Timp3* in EpiSV40 cells (n = 3). *Actb* was a negative control. Data information: in (C, F, G, H, K, L), data are presented as mean ± SEM. NS, not significant; ∗P < 0.05, ∗∗P < 0.01, and ∗∗∗P < 0.001.

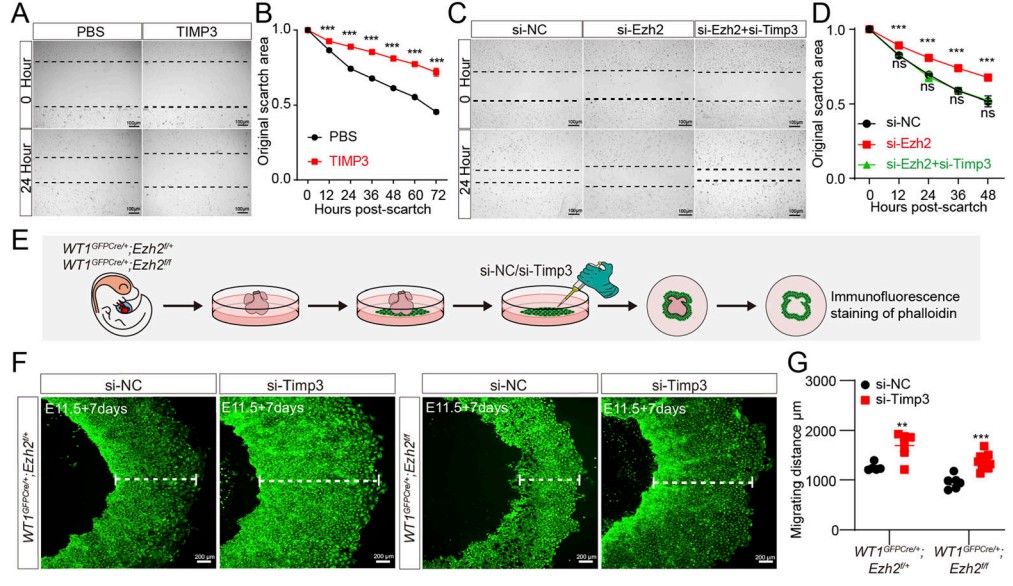

**Figure 5. TIMP3 is implicated in EZH2-induced epicardial cell migration.**

**(A)** Scratch assay following TIMP3 recombinant protein treatment. Scale bar, 100 μm. **(B)** Migration in (A) is presented as a percentage of the original scratch area. (n = 5 for each condition). **(C)** Scratch assay following knockdown of *Ezh2* and combined knockdown of *Ezh2* and *Timp3*. Scale bar, 100 μm. **(D)** Migration in (C) is presented as a percentage of the original scratch area. (n = 5 for each condition). **(E)** Primary epicardial cells were isolated from E12.5 *Ezh2^epi-KO^* mice, and *Timp3* knockdown was performed with siRNA. **(F)** Immunostaining for phalloidin (green) in *Ezh2^epi-KO^* mouse epicardial explants. Scale bar, 200 μm. **(G)** Migration in (F) is presented as cell migration width (n = 6 for each condition). Data information: in (B, D, G), data are presented as mean ± SEM. NS, not significant; ∗P < 0.05, ∗∗P < 0.01, and ∗∗∗P < 0.001.

et al, 2017). To gain insight into TIMP3-dependent ECM reconstruction involved in epicardial cell migration, we extracted ECM from E12.5 *Ezh2^epi-KO^* hearts and conducted a proteomic analysis by tandem mass spectrometry (MS/MS) (Fig 6A). Gene set enrichment analysis (GSEA) revealed that ECM-receptor interaction was most significantly up-regulated in *Ezh2^epi-KO^* hearts (Fig 6B), which indicates that epicardial *Ezh2* deletion blocked ECM degradation.

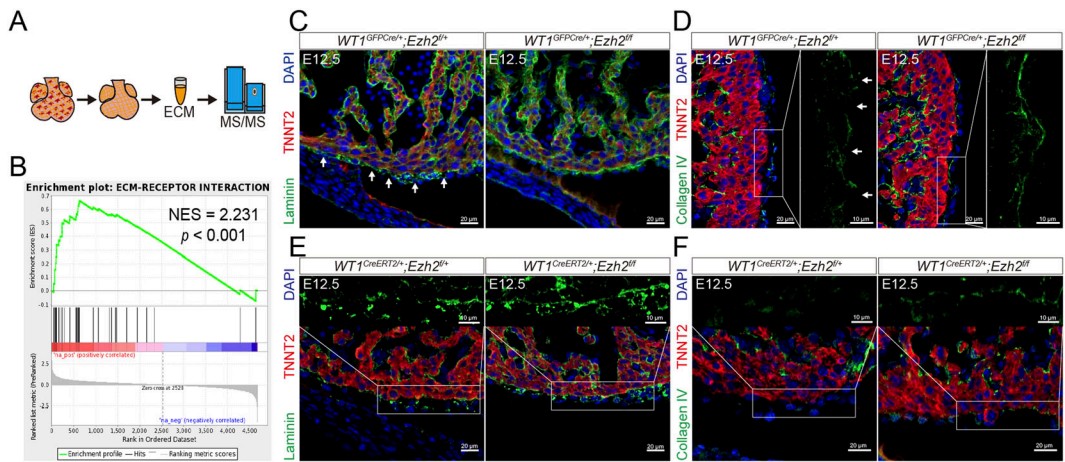

**Figure 6. EZH2/TIMP3 regulates epicardial cell migration by activating ECM degradation.**
**(A)** ECM extracted from E12.5 *Ezh2^epi-KO* hearts was subjected to proteomic analysis by MS/MS. **(B)** GSEA enrichment plot for the ECM-receptor interaction gene set.
**(C)** *Ezh2^epi-KO* mouse hearts displayed an intact basement membrane according to laminin (green) immunohistochemistry. Scale bar, 20 μm. **(D)** *Ezh2^epi-KO* mouse hearts displayed an intact basement membrane according to collagen IV (green) immunohistochemistry. Scale bar, 20 μm, 10 μm. **(E)** *Ezh2^epi-iKO* mice hearts displayed an intact basement membrane in laminin (green) immunohistochemistry. Scale bar, 20 μm, 10 μm. **(F)** *Ezh2^epi-iKO* mice hearts displayed an intact basement membrane in collagen IV (green) immunohistochemistry. Scale bar, 20 μm, 10 μm.

Among the cardiac ECM components influenced in *Ezh2^epi-KO* hearts, laminin and collagen IV are the major components of the basement membrane. Epicardial cell migration into the myocardium through the basement membrane is essential for cardiac development, which is attributed to digestion of the ECM (Sun et al, 2021). Immunostaining revealed that epicardial cell–specific *Ezh2* deletion caused an intact basement membrane, which was shown as the *Ezh2^epi-KO* and *Ezh2^epi-iKO* mouse hearts displayed an intact basement membrane, although the membrane was disrupted in *WT1^GFPCre/+;Ezh2^flox/+* hearts and *WT1^CreERT2/+;Ezh2^flox/+* hearts (Fig 6C–F).

Taken together, our results suggested that epicardial deletion of *Ezh2* inhibited basement membrane degradation, resulting in defects in epicardial cell migration.

## Discussion

In this study, we revealed that EZH2 was expressed in human and mouse epicardial cells during cardiac development. *Ezh2* epicardial deletion resulted in myocardial hypoplasia and defective coronary plexus development, leading to embryonic lethality. Furthermore, we deciphered the mechanisms by which EZH2 participates in epicardial development, regulating epicardial cell migration and degradation of the basement membrane by suppressing the transcription of *Timp3*. Overall, we demonstrated that the EZH2–TIMP3 system regulates epicardial cell migration, which provides new insight into the function of EZH2 in cardiac development.

As a histone methyltransferase, EZH2 has been reported to play critical roles in cardiac development and heart regeneration in terms of regulating cardiomyocyte proliferation and maintenance (Chen et al, 2012; Delgado-Olguín et al, 2012; He et al, 2012; Yue et al, 2019). Myocardial inactivation of EZH2 by *Nkx2.5*-Cre or *Myh6*-CreERT2 leads to cardiomyocyte proliferation arrest (He et al, 2012; Yue et al, 2019). In

addition, *Mef2c*-Cre-induced *Ezh2* deletion in cardiac progenitor cells results in postnatal myocardial pathology (Delgado-Olguín et al, 2012). Our results showed that EZH2 and H3K27me3 were present in both the human and mouse embryonic epicardium, indicating the conserved function of EZH2 in epicardium development. *Ezh2* deficiency resulted in epicardial cell migration inhibition and cardiac malformation. Substantial evidence has demonstrated that EZH2 participates in embryonic development by regulating cell proliferation and differentiation. Here, we found that EZH2 could also be a cell migration regulator in epicardial development.

Migration disorder of cardiac progenitor cells and neural crest cells leads to a spectrum of cardiac outflow tract defects (Epstein et al, 2000; Cai et al, 2003; Ramsbottom et al, 2014; Bahm et al, 2017). Epicardial cell migration defects result in impaired coronary vasculature development and ventricular dysplasia (Singh et al, 2016; Liu et al, 2018; Xiao et al, 2018). Our results revealed that EZH2 regulated epicardial cell migration via reconstruction of the cardiac ECM. By using RNA-Seq, we unbiasedly found that *Ezh2* deficiency promoted the expression of *Timp3*, which is attributed to the ECM degradation process and is essential for heart development (Ratajska & Cleutjens, 2002; Zhang et al, 2013). A previous study found that mouse embryos with epicardial malformations caused by WT1 mutations would have ventricular dysplasia, especially with the thinning of the dense ventricular layer. This suggests a relationship between the adjacent epicardium and the myocardium, whose proliferation and development may be regulated by the epicardium and paracrine effect (Duim et al, 2016). In our study, GO analysis showed that genes were enriched in terms of cell migration, which showed no signs of paracrine signal regulation in *Ezh2*-deficient epicardial cells.

The basement membrane acts as a barrier, and its disruption might facilitate the migration of epicardial cells into the myocardium (Baek & Tallquist, 2012; Sun et al, 2021). We revealed that epicardial *Ezh2* deletion blocked basement membrane degradation and resulted in epicardial cell migration disorder. ECM mass

spectrometry revealed that the levels of the components of the basement membrane, laminin, and collagen IV, were increased in epicardial *Ezh2*–deficient mice. Therefore, the basement membrane is involved in epicardial cell migration and epicardial development during heart development, which is regulated by the EZH2–TIMP3 system. We believe that epicardial *Ezh2* deficiency disturbs multiple factors and cell behaviors, leading to cardiac malformation and fetal lethality. Our results suggest that *Ezh2* knockdown affects epicardial cell proliferation; we do not rule out the effect of *Ezh2*-specific knockout on the proliferation ability of epicardial cells. However, the novel finding in this study is that *Ezh2* deletion regulates epicardial cell migration during embryonic heart development. Further efforts will be made to elucidate more effects of EZH2 on epicardial cells.

In summary, our study demonstrates the role of EZH2 in epicardial cell migration during heart development. We reveal that the interaction of epicardial cells with the basement membrane is fine-tuned by the EZH2–TIMP3 system during heart development. These findings broaden the understanding of EZH2, facilitating cell migration via regulation of ECM. Our results provide new insights into the role of epicardial cells in heart development, which control the integrity of the basement membrane and epicardial cell migration into the myocardium.

# Materials and Methods

## Human embryos

Human embryos were obtained after voluntarily terminated pregnancies with the parent's written informed consent (Haidian Maternal & Child Health Hospital, Beijing, China). All experiments were conducted in accordance with protocols evaluated and approved by the Peking University Institutional Review Board (PU-IRB) (certificate number: IRB00001052-18083).

## Mice

Epicardium-specific *Ezh2* knockout mice were generated by crossing the *WT1^{GFPCre/+}* with *Ezh2^{flox/flox}* mice. We constructed inducible *Ezh2* epicardial conditional knockout (cKO) mice by crossing *WT1^{CreERT2/+}* mouse line with *Ezh2^{flox/flox}*. Two doses of tamoxifen were administered at E9.5 and E10.5. The *WT1^{GFPCre/+}* and *WT1^{CreERT2/+}* mice lines were kindly provided by Dr. Bin Zhou (Zhou et al, 2008). The *Ezh2^{flox/flox}* mice line was obtained from the Jackson Laboratory (strain name *Ezh2tm2Sho*, stock number 022616). *Ezh2^{flox/flox}* animals were genotyped as described previously (Yue et al, 2019). The wild-type C57BL/6J pregnant mice were obtained from SPF (Beijing) Biotechnology Co., Ltd. for experiments. For timed pregnancies, midday of vaginal plug was considered embryonic day 0.5 (E0.5). All experiments involving animals were conducted in accordance with the Guide for the Use and Care of Laboratory Animals. All animal protocols were approved by the Institutional Animal Care and Use Committee (IACUC), Fuwai Hospital, Chinese Academy of Medical Sciences.

## Histology and immunohistochemistry

Histology and immunohistochemistry were performed as described previously (Chu et al, 2020). Briefly, mouse embryos were dissected in PBS and fixed in 4% PFA overnight at 4°C. The embryos were washed with PBS, dehydrated in ethanol, and embedded in paraffin. Sections of 5 *µ*m in thickness were deparaffinized in xylene and rehydrated through a graded ethanol series (100%, 95%, 75%, and 50%) and then stained for hematoxylin & eosin (H&E) staining or immunohistochemical detection. Immunohistochemical detection was performed on paraffin sections of 4% PFA-fixed hearts. Heat-mediated antigen retrieval using EDTA solution was applied to the sections. For immunohistochemical detection of cell explants, cell explants were washed in PBS and fixed in 4% PFA for 10 min at 4°C. Then, tissue sections or cell explant chambers were blocked in 5% donkey serum (Invitrogen) with 0.3% Triton X-100 (Sigma) and incubated with primary antibodies overnight at 4°C. Each sample was washed for 5 min five times with PBS, and thereafter, the samples were incubated with secondary antibodies conjugated to Alexa Fluor 488/594/647 (Invitrogen) for 1 h at 37°C. Fluorescence was observed under a confocal laser scanning microscope (LSM800; ZEISS). The primary antibodies used for immunohistochemistry were anti-EZH2 rabbit polyclonal (Cat. no. 39933; Active Motif), anti-H3K27me3 rabbit polyclonal (Cat. no. 07-449; Millipore), anti-Tnnt2 mouse monoclonal (Cat. no. ab8295; Abcam), anti-WT1 rabbit polyclonal (Cat. no. ab89901; Abcam), anti-GFP goat polyclonal (Cat. no. NB100-1770SS; Novus), anti-CD31 rabbit polyclonal (Cat. no. ab32457; Abcam), anti-CDH5 goat polyclonal (Cat. no. AF1002; Novus), anti-phospho-histone H3 (pH3) rabbit polyclonal (Cat. no. 06-570; Millipore), anti-Ki67 rat monoclonal (Cat. no. 14-5698-82; Invitrogen), anti-laminin rabbit polyclonal (Cat. no. ab11575; Abcam), and anti-collagen IV rabbit polyclonal (Cat. no. ab6585; Abcam) antibodies. A high-content screening system (Opera Phenix) was used to count the cell proliferation ratio.

Whole-mount immunostaining for PECAM1 was carried out as described previously (Liu et al, 2018). Briefly, samples were fixed in methanol/DMSO (4:1) overnight at 4°C and then bleached in methanol/DMSO/30% hydrogen peroxide (4:1:1) for 4 h at RT. After rehydration through 50% methanol, 15% methanol, and PBT-3 (PBS containing 0.3% Triton X-100), the samples were incubated with an anti-rabbit PECAM1 polyclonal antibody (Cat. no. ab32457; Abcam) at a 1:800 dilution in PBTG (PBS containing 0.3% Triton X-100 and 10% goat serum) at 4°C overnight. Next, samples were washed twice in PBT-3 and twice in PBT-5 (PBS containing 0.5% Triton X-100) for 30 min each. After washing, the secondary antibody HRP-conjugated goat anti-rabbit IgG was applied overnight at 4°C at a dilution of 1:500, and color development was performed using a DAB kit (Cat. no. ZLI-9018; ZSGB-BIO). The area of vessel coverage was calculated as a percentage (%) of the PECAM1⁺ plexus surface area covering surface area of the dorsal side of the heart.

## RNAscope assay

RNA in situ hybridization (ISH) was performed using the RNAscope. The RNAscope reagent kit (Cat. No. 323100; Bio-Techne) and RNAscope assay combined with the Immunohistochemistry–Integrated Co-Detection Reagent Kit (Cat. No. 323180; Bio-Techne) were used

according to the manufacturer's instructions. Briefly, paraffin sections of 5 μm in thickness were deparaffinized in xylene, rehydrated through 100% ethanol, and then air-dried at RT. The slices in hydrogen peroxide were incubated for 10 min at RT. The slide rack was very slowly submerged with a pair of forceps into the 98–102°C co-detection target retrieval solution for 20 min. Immediately, the hot slide rack was transferred to a staining dish containing distilled water. Each sample was washed for 2 min two times with PBST, and thereafter, the samples were incubated with the anti-EZH2 rabbit polyclonal antibody overnight at 4°C. Place slides in 10% neutral buffered formalin for 30 min at RT. Slides were washed in PBST, and then two to four drops of RNAscope Protease Plus was added to each section. The HybEZ Slide Rack was placed in the pre-warmed HybEZ oven (Bio-Techne Co.) and incubated at 40°C for 30 min. After protease plus incubation, the sections were washed with distilled water before being subjected to an RNAscope hybridization assay. The probes were designed and produced by Bio-Techne and depended on the targets' FASTA format nucleotide sequences provided on the NCBI database. To start the hybridization, the RNA probe fluid was warmed in a 40°C water bath for 10 min, and then cooled down to RT. The probe was added to the sections, and incubation was carried out in an HybEZ oven for 2 h at 40°C. After hybridization incubation, the slides were washed using 1x RNAscope wash buffer. Then, sections were incubated with RNAscope Multiplex FL v2 Amp 1, Amp 2, and Amp 3 (for 30/30/15 min, respectively) sequentially at 40°C to amplify the signal. For signal development, RNAscope Multiplex FL v2 HRP-C1 (for lnc-mg) was added to the sections (incubation time 15 min, 40°C). For revealing signals, TSA-diluted Opal 590 fluorophore was added to sections, incubating sections for 30 min at 40°C. After fluorophore incubation and rinse with 1× RNAscope wash buffer, RNAscope Multiplex FL v2 HRP blocker was added and incubated in the oven at 40°C for 15 min. Sections were washed using 1× RNAscope wash buffer. Secondary antibody diluted in co-detection antibody diluent was added to the sections and incubated for 30–60 min at RT. Enough solution was used to completely cover the sections. Slides were washed with PBST for 2 min two times. Finally, the sections were counterstained with RNAscope DAPI (30 s), and the slides were cover-slipped with ProLong Glass Antifade Mountant (Thermo Fisher Scientific). Fluorescence was observed under a confocal laser scanning microscope (LSM800; ZEISS).

## EdU incorporation

EdU was used to label epicardial cells undergoing mitosis in vitro and in vivo. Cells grown on coverslips were pretreated with 10 μM EdU for 2 h in DMEM with 10% fetal bovine serum and subjected to EdU immunostaining. For in vivo labeling, EdU was diluted in PBS and intraperitoneally injected into the pregnant mice (10 mg/kg) 4 h before embryonic heart collection. After dissection, embryos or hearts were fixed with 4% PFA and embedded in paraffin. EdU incorporation was detected using BeyoClick EdU cell proliferation kit with Alexa Fluor 488 (Beyotime, Cat. no. C0071S). Tissue slides were imaged with a confocal laser scanning microscope (LSM800; ZEISS). A high-content screening system (Opera Phenix) was used to count cell proliferation ratio.

## Epicardial cell culture

Primary epicardial cells were isolated from *Ezh2^epi-KO* embryos as described (Jiang et al, 2021). The hearts dissected from E11.5 embryos were placed on a 0.1% gelatin gel. The epicardial cells migrated onto the dish and formed an epithelial monolayer after 2 d in epicardial culture medium DMEM supplemented with 100 units/ml penicillin, 100 μg/ml streptomycin, 10% FBS, and 1% glutamine. The hearts were then removed, and the explants were cultured for another 3–5 d. Epicardial cells were harvested in TRIzol (Cat. no. 15596-018; Life Technologies) for RNA isolation or cultured for further analysis. siRNA transfections were performed according to standard protocols using DharmaFECT transfection reagent (Dharmacon). Epicardial cells were transfected with 25 nM siRNA SMARTpools (Dharmacon ON-TARGET plus) against *Ezh2* (LQ-040882–00; Dharmacon), *Timp3* (LQ-040739–01; Dharmacon), or nontargeting siRNA (D-001810–01; Dharmacon). Recombinant human TIMP3 was purchased from R&D (973-TM-010), and 200 ng/ml was used to treat epicardial cells.

## Western blotting assay

The epicardial cells were lysed in RIPA lysis buffer with 1 mM PMSF (Beyotime Institute of Biotechnology). After homogenization and incubation at 4°C for 30 min, all protein samples were mixed with 1:4 4 × SDS-loading buffer and 1:10 10X SDS-loading buffer and boiled for 10 min at 95°C. Subsequently, 10 μg of total protein was loaded onto SDS–PAGE and subsequently blotted onto a nitrocellulose membrane. After blocking the nonspecific background staining, the membranes were incubated at 4°C overnight with primary antibodies as follows: TIMP3 (ab276134, 1:1,000; Abcam), EZH2 (Cat. no. 39933, 1:1,000; Active Motif), and GAPDH (8884, 1:5,000; Cell Signaling Technology) were used as controls. Then, the membranes were washed with Tris-buffered saline containing 0.1% Tween 20 and incubated with the secondary antibody for 1 h at RT. Signals were detected using Pierce ECL Western Blotting Substrate (32209; Thermo Fisher Scientific).

## RNA sequencing

After being treated with si-NC or si-Ezh2 for 48 h, epicardial cells were harvested in TRIzol. The quantity and quality of total RNA were measured with a NanoDrop 2000 and Agilent Bioanalyzer 2100, respectively. Total RNA (1 μg) with an RNA integrity number (RIN) value of no less than 8.0 for each sample was used for library preparation. cDNA libraries were prepared and sequenced on an Illumina MiSeq using the paired-end run methodology at Novogene. The raw reads were generated, and quality control determination was performed using standard protocols. RNA-Seq reads were aligned to the mouse genome by HISAT2 (Version 2.0.5), and the gene expression level was quantified as fragments per kilobase of exon per million fragments (FPKM) using featureCounts (Version 1.5.0-p3). DESeq2 (Version 1.16.1) was used to determine the genes that were differentially expressed between the si-NC and si-Ezh2 groups. Genes with an FPKM > 1, a fold change > 2, and a false discovery rate < 5% were considered to be differentially expressed. GO enrichment analysis was performed using String v9.0 (http://string-db.org). Variables with adjusted *P*-values < 0.05 were considered significantly enriched. The raw sequence data reported in this study have been deposited in the

**Table 1. Primer sequences for qRT-PCR.**

| Gene | Primer sequences |
|------|------------------|
| Wt1 | F: 5′-GAGAGCCAGCCTACCATCC-3′;<br>R: 5′-GGGTCCTCGTGTTTGAAGGAA-3′ |
| Raldh2 | F: 5′-GGCACTGTGTGGATCAACTG-3′;<br>R: 5′-TCACTTCTGTGTACGCCTGC-3′ |
| Nkx2.5 | F: 5′-GACGTAGCCTGGTGTCTCG-3′;<br>R: 5′-GTGTGGAATCCGTCGAAAGT-3′ |
| Myh6 | F: 5′-ATAAAGGGGCTGGAGCACTG-3′;<br>R: 5′-GCCTCTAGGCGTTCCTTCTC-3′ |
| Myh7 | F: 5′-GTGGCTCCGAGAAAGGAAG-3′;<br>R: 5′-GAGCCTTGGATTCTCAAACG-3′ |
| Tnnt2 | F: 5′-ATTCGCAATGAGCGGGAGAA-3′;<br>R: 5′-ACCCTCCAAAGTGCATCATGT-3′ |
| Ezh2 | F: 5′-TTGTGACAGTTCGTGCCCTTG-3′;<br>R: 5′-GTAGCATGGACACTGTTTGGTGTTG-3′ |
| Timp3 | F: 5′-AGGAGTGGGTCTCACAGTTATCC-3′;<br>R: 5′-ACGTTCATCTCAGCCCTTTGA-3′ |
| Actb | F: 5′-GGTACCACCATGTACCCAGG-3′;<br>R: 5′-AAAACGCAGCTCAGTAACAGTC-3′ |

Genome Sequence Archive (GSA) under accession number CRA003959.

### Quantitative reverse transcription–polymerase chain reaction (qRT-PCR)

Total RNA was isolated from epicardial explants using TRIzol. PrimeScript RT Master Mix was used to convert RNA into cDNA. qRT-PCR for analysis of the expression of different genes (Table 1) was performed in triplicate using SYBR Green qRT-PCR Master Mix (Applied Biosystems) in a Vii7 Real-Time PCR System (Applied Biosystems). The results were analyzed using GraphPad Prism, and differences between groups were analyzed using $t$ test. The significance levels were indicated by Prism software as follows: $*P < 0.05$, $**P < 0.01$, $***P < 0.001$.

### Scratch migration assay

Cells transfected for 48 h with siRNAs were plated in 24 well plates and cultured to confluency. Then, epicardial cells were serum-starved and scraped with a 10 $\mu$l pipette tip (0 h), washed with PBS, and cultured with serum-free DMEM. Images were taken at the indicated time points.

### ChIP assay

ChIP assays were performed using a ChIP Kit (Cat. no. #9003; CST) according to the manufacturer's instructions. Briefly, cells were cross-linked with formaldehyde and sonicated to an average length of 200–1,000 bp. Immunoprecipitation was conducted against H3K27me3 (Cat. no. #9733; CST), Suz12 (Cat. no. #3737; CST), or IgG control. The immunoprecipitated DNA was quantified by RT-PCR in a

Vii7 Real-Time PCR System (Applied Biosystems). The primer sequences are as follows:

Timp3-1 kb
F: 5′-ACAAGTTGTAGTCTGTGCGGG-3′,
R: 5′-GGCGCGGGAGATAAGCAATC-3′;
Timp3-1.3 kb
F: 5′-GATAGCTGGCCTGAGGTGAC-3′,
R: 5′-CTCACAAACATTCCGGGGGA-3′;
Timp3-1.6 kb
F: 5′-TGCCAAAGGTCTCATCGCTT-3′,
R: 5′-TCCCCAACACTCCTCTGGAT-3′;
Actb
F: 5′-CGTATTAGGTCCATCTTGAGAGTACACAGTATT-3′,
R: 5′-GCCATTGAGGCGTGATCGTAGC-3′.

### Proteomic analysis

For each proteomic sample, five cardiac ECM samples were pooled together. Cardiac ECM extraction was performed as described previously (Bassat et al, 2017). Briefly, hearts were collected from E12.5 $Ezh2^{epi-KO}$ and $WT1^{GFPCre/+};Ezh2^{flox/+}$ mice, washed with PBS, and embedded in OCT compound at −20°C. The OCT embedded hearts were cut transversely into 50 $\mu$m fragments and immersed in 2% Triton X-100 and 20 mM EDTA solution in double-distilled water overnight at RT. ECM was dissolved in lysate (8 M urea, 0.1 M Tris–HCl, pH 8.5) and digested into peptides by trypsin. After trypsin digestion, the peptide was desalted with a Strata X C18 SPE column (Phenomenex) and vacuum dried. The peptide was reconstituted in 0.5 M TEAB and processed according to the manufacturer's protocol for the TMT kit/iTRAQ kit. The peptides were subjected to a nanospray ionization (NSI) source followed by tandem mass spectrometry (MS/MS) in a Q Exactive Plus (Thermo) coupled online to the ultrahigh performance liquid chromatography (UPLC). The resulting MS/MS data were processed using the MaxQuant search engine (v.1.5.2.8). The false discovery rate was adjusted to < 1%, and the minimum score for modified peptides was set to > 40. GSEA was performed with GSEA software (www.broadinstitute.org/gsea). Variables with adjusted $P$ values < 0.05 were considered significantly enriched.

### Statistical analysis

All results were analyzed using GraphPad Prism (version 7.0; GraphPad Software). All data are expressed as the mean ± SEM. Differences between two groups were evaluated using an unpaired $t$ test. One-way ANOVA was performed for more than two group comparisons, and two-way mixed-effects ANOVA was performed for multiple group comparisons at different time points. The results with $P$-values < 0.05 were considered statistically significant.

## Data Availability

The RNA-seq datasets generated during the current study have been deposited and are available from Genome

Sequence Archive (GSA) repository with the accession code CRA003959.

## Supplementary Information

## Acknowledgements

We sincerely thank Jian Meng (State Key Laboratory of Cardiovascular Disease, Fuwai Hospital) for technical support with the tissue slices and immunochemistry. This work was supported by the National Key Research and Development Project of China (2019YFA0801500), the Chinese Academy of Medical Sciences (CAMS) Innovation Fund for Medical Sciences (CAMS-I2M, 2021-I2M-1-072), the National Natural Science Foundation of China (81770308, 81900343), and the Chinese Academy of Medical Sciences Innovation Fund for Medical Sciences (CIFMS, 2016-I2M-1-015).

### Author Contributions

H Jiang: conceptualization, data curation, software, formal analysis, visualization, methodology, and writing—original draft, review, and editing.
L Bai: conceptualization, data curation, formal analysis, validation, investigation, visualization, methodology, and writing—original draft, review, and editing.
S song: conceptualization, methodology, and writing—original draft.
Q Yin: conceptualization, resources, and software.
A Shi: resources and software.
B Zhou and H Chen: conceptualization and writing—review and editing.
H Lian: data curation, methodology, and writing—review and editing.
C-r Xu: methodology and writing—review and editing.
Y Wang: resources and writing—review and editing.
Y Nie: conceptualization, resources, data curation, supervision, funding acquisition, visualization, project administration, and writing—original draft, review, and editing.
S Hu: conceptualization, resources, data curation, formal analysis, supervision, funding acquisition, visualization, project administration, and writing—review and editing.

### Conflict of Interest Statement

The authors declare that they have no conflict of interest.

### Ethics statement

The authors declare no competing interests. All experiments were conducted in accordance with protocols evaluated and approved by the Peking University Institutional Review Board (PU-IRB) (certificate number: IRB00001052-18083). All experiments involving animals were performed and approved by the Institutional Animal Care and Use Committee (IACUC), Fuwai Hospital, Chinese Academy of Medical Sciences.

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
