## [Reviewer comments · Life Science Alliance]

Life Science Alliance

EZH2 controls epicardial cell migration during heart development

Haobin Jiang, Lina Bai, Shen song, Qianqian Yin, Anteng Shi, Bin Zhou, Hong Lian, Hou-Zao Chen, Cheng-Ran Xu, Yanchun Wang, Yu Nie, and Shengshou Hu

DOI: <https://doi.org/10.26508/lsa.202201765>

Corresponding author(s): Yu Nie, State Key Laboratory of Cardiovascular Disease, Fuwai Hospital, National Center for Cardiovascular Disease and Yu Nie, State Key Laboratory of Cardiovascular Disease, Fuwai Hospital, National Center for Cardiovascular Disease

Review Timeline:

Submission Date:	2022-10-11
Editorial Decision:	2022-11-07
Revision Received:	2023-02-28
Editorial Decision:	2023-03-15
Revision Received:	2023-03-26
Accepted:	2023-03-27

Scientific Editor: Novella Guidi

Transaction Report:

November 7, 2022

Re: Life Science Alliance manuscript #LSA-2022-01765-T

Dr. Yu Nie
State Key Laboratory of Cardiovascular Disease, Fuwai Hospital, National Center for Cardiovascular Disease, Chinese Academy of Medical Sciences and Peking Union Medical College
State Key Laboratory of Cardiovascular Disease
Beijing, Please Select 100037
China

Dear Dr. Nie,

Thank you for submitting your manuscript entitled "EZH2 controls epicardial cell migration during heart development" to Life Science Alliance. The manuscript was assessed by expert reviewers, whose comments are appended to this letter. We invite you to submit a revised manuscript addressing the Reviewer comments.

Thank you for this interesting contribution to Life Science Alliance. We are looking forward to receiving your revised manuscript.

Sincerely,

B. MANUSCRIPT ORGANIZATION AND FORMATTING:

Reviewer #1 (Comments to the Authors (Required)):

This manuscript describes the function of EZH2 in epicardial cells during heart development. By deleting EZH2 with WT1-GFP-Cre, the authors investigate the heart phenotypes and claim that EZH2 is essential for epicardial cell migration into myocardium and that EZH2 suppresses Timp3 transcription, which controls the integrity of basement membrane and therefore migration of epicardial cells.

Even though there are merits in this manuscript, this manuscript does not appear to present convincing evidence to support its claim. Major concerns are:

1. No in vivo epicardial cell migration data are present. Epicardial cell migration from epicardium to heart during development is a well-established event during heart development. GFP reporter in WT1-GFP-Cre; EZH2 f/f would seem to be an ideal tool to determine the effect of EZH2 deletion on the epicardial cell migration in vivo.
2. Epicardial migration but not proliferation/anti-cell death is claimed to be the major role of EZH2. However, the data to support this claim are very weak. Again, no in vivo data and only weak in vitro data. Careful migration, proliferation and cell death assays are needed.
3. Timp3 as a possible target of EZH2 is interesting. Yet only Timp3 expression data is present in the manuscript. To make a rather solid conclusion, evidence to show that EZH2 binds to Timp3 promoter and changes histone acetylation/other modifications would be minimally required.
4. The claim that EZH2 is involved in human epicardial development also lacks real data support. The presence of its expression in human epicardial cells is only a piece of weak association evidence.

Reviewer #2 (Comments to the Authors (Required)):

In this study, the investigators have applied immunostaining, qRT-PCR, scratch assay and loss-of-function studies on epicardial cells and found EZH2 is required for mice embryonic survival and epicardial cell migration. The authors then further determined EZH2 influenced epicardial cell migration through regulating ECM components by RNA-seq, ECM mass spectrometry and ChIP-qPCR. Overall, this is a clearly written manuscript with experimental results supporting most of the findings. The reviewer only has several minor concerns.

Figure 2B: Please consider to show a survival curve of the EZH2^{epi}-ko embryos only instead of the bar chart of each genotype, since the figure legend shows "survival of EZH2^{epi}-ko embryos". The ratio of other genotypes can be included in the supplementary data.

Figure 2C: Did the authors examine EZH2 expression in EZH2^{epi}-ko embryos at earlier stages, like E10.5? What is the knockout efficiency would be? At what stage does WT1GFP^{cre} become active?

Figure 3G: For the RNA-seq, is the 1ug of total RNA obtained from a single heart or a pool of several hearts?

Figure 4B: Is there any difference in Timp3 protein expression level between WT and EZH2^{epi}-ko embryos at E12.5? It can be done by immunostaining of Timp3 antibody.

Figure 4L/M: How many cells were used for each ChIP? Panel M, is the average number of Actb (negative control) less than 1? Why the sample size for ChIP-qPCR is not consistent?

Reviewer #3 (Comments to the Authors (Required)):

In this manuscript, the authors investigated the role of Ezh2 in the developing epicardium. They use Wt1 lineage-specific deletion of Ezh2 to reveal an essential role during embryonic heart development and specifically suggest that epicardial cell migration is compromised. They reveal that Timp3 is upregulated in Ezh2KO epicardial cells, whereas enhanced Timp3 prohibits extracellular matrix degradation and compromises epicardial cell migration. This work is of interest and extends the understanding of Ezh2 in heart development, specifically acting on the epicardium. However, there are some concerns with the study in its current form that ought to be addressed:

- 1) In Figure 1, the authors detected Ezh2, H3K27me3 and WT1 expression in human and mouse embryonic epicardium with immunostaining. However, the authors did not show a correlation between Ezh2 and H3K27me3 in epicardial cells. Wt1 is the only marker used to label epicardium. To prove the expression of Ezh2 and H3K27me3 expression/loss in epicardial cells further epicardial markers should be used.
- 2) Wt1 is expressed in the intermediate mesenchyme from E9.0 and is also expressed in the coronary endothelium (Armstrong JF et al., 1992). Thus, the phenotype of Ezh2 deletion in Wt1-Cre is not only due to Ezh2 loss in the epicardial lineage but also other cell types, such as endothelial cells. For example, the defects shown in Fig.2D, H, L and K could be the direct consequence of Ezh2 loss in CD31+ cells rather than Ezh2 loss in epicardial cells. To exclude this possibility, the authors should use an inducible Cre, such as Wt1-CreErt2 (Zhou B et al., 2008; Zhou B and Pu W, 2012) to delete Ezh2 in the epicardium more specifically.
- 3) Aside from the reported migration defects, the authors have not characterized in sufficient depth the epicardial defects of Ezh2KO mutants. For example, the authors should evaluate epicardial cell proliferation and quantify WT1+ cells in the epicardium and WT1+ EPDCs in the myocardium; alongside further epicardial markers such as podoplanin or Integrin alpha4/beta1.
- 4) The authors detected Timp3 upregulation based on RNA profiling, however, they should also determine whether Timp3 is upregulated in Ezh2 knockout epicardium using an in situ RNA-based approach (spatial HCR, RNAscope) and at the level of the protein via immunostaining.
- 5) The authors suggest that the absence of Ezh2 results in decrease of H3K27me3 at the promotor of Timp3, however, Figures 3 and 4 do not address whether this is cause or effect on the altered Timp3 expression in the KOs, which needs addressing. The authors should also include whether Timp3 is detected in the MS analysis and GSEA (Figure 4N and 4O).

Reviewer #4 (Comments to the Authors (Required)):

In this elegantly designed study by Jiang et al demonstrates a crucial role of EZH2 in epicardial cell migration and cardiac development. The data are high quality and properly interpreted, and support the main conclusion in general. Overall, this is a nice manuscript with interesting findings; however, a few questions should be clarified.

Specific comments:

1. In all figures, the embryos of WT1GFP^{Cre/+};EZH2^{f/+} was used as controls which exhibited normal epicardial cell migration and heart development, suggesting 50% of EZH2 is sufficient to maintain the proper function of epicardial cells. In contrast, 50% reduction of EZH2 in cultured epicardial cells showed dramatic phenotype. This discrepancy should be explained.
2. Since EZH2 is essential for cell proliferation of cardiomyocyte as introduced at begin, it will be interesting to check whether reduced cell proliferation contributed to the cardiac phenotype?
3. Mechanistically, the authors showed that knockdown of EZH2 in cultured epicardial cells upregulated the expression of TIMP3. Consistently, Laminin and Collagen IV, two targets of TIMP3, were increased in the EZH2 KO hearts. The authors concluded that EZH2-dependent suppression of TIMP3 is essential for the degradation of ECM and epicardial cell migration. Since TIMP3 is the key downstream mediator of EZH2, the in vivo data is needed to show the upregulation of TIMP3 in EZH2 knockout hearts.
4. The deletion of EZH2 in epicardium severely impeded the myocardial growth, suggesting EZH2-dependent paracrine signals in epicardium is essential for myocardial development. However, this was not studied or discussed in the paper.
5. The images in Figure4P,Q showed that the expression of Laminin and Collagen IV was increased in both sub-epicardium and sub-endocardium of mutant embryos, whereas the deletion is epicardial-specific, suggesting that the upregulation of those ECM proteins may not be specifically due to the loss of EZH2.

Minor point: The front size in Figure 4O is too small.

To Reviewers:

Reviewer #1: This manuscript describes the function of EZH2 in epicardial cells during heart development. By deleting EZH2 with WT1-GFP-Cre, the authors investigate the heart phenotypes and claim that EZH2 is essential for epicardial cell migration into myocardium and that EZH2 suppresses Timp3 transcription, which controls the integrity of basement membrane and therefore migration of epicardial cells.

Even though there are merits in this manuscript, this manuscript does not appear to present convincing evidence to support its claim. Major concerns are:

1.No *in vivo* epicardia cell migration data are present. Epicardial cell migration from epicardium to heart during development is a well-established event during heart development. GFP reporter in WT1-GFP-Cre;EZH2 *f/f* would seem to be an ideal tool to determine the effect of EZH2 deletion on the epicardial cell migration *in vivo*.

Response:

Thank you very much for your comments. In this study, we crossed EZH2^{f/f} mice with the WT1^{GFP-Cre/+} murine line to generate an epicardium-specific EZH2 loss-of-function mutant. Our *in vivo* data of immunofluorescence staining of WT1 showed that EPDCs were undetectable in the compact myocardium of E12.5 EZH2^{epi-KO} hearts, while EPDCs existed in WT1^{GFP-Cre/+};EZH2^{f/+} hearts (Figure R1A), which indicated EZH2 involves in epicardial cell migration. Next, to explore the role of EZH2 in regulating epicardial cell migration *in vivo*, we performed explant heart culture and found a significantly decreased outgrowth of epicardial cells in EZH2^{epi-KO} heart explants relative to WT1^{GFP-Cre/+};EZH2^{f/+} hearts (Figure R1B, C).

Besides, we constructed inducible EZH2 epicardial conditional knockout (cKO) mice by crossing WT1^{CreERT2/+} mouse line with EZH2^{f/f}. Two doses of Tamoxifen were administered at E9.5 and E10.5, WT1^{CreERT2/+};EZH2^{f/f} (hereafter as EZH2^{epi-iKO}) epicardial cells exhibited markedly reduced EZH2 immunoreactivity by E12.5 (Figure R1D, E). EZH2^{epi-iKO} mouse could survive to E15.5. This cardiac phenotype of

$EZH2^{epi-iKO}$ is generally consistent with that of $EZH2^{epi-KO}$, which showed decreased epicardial cell number, and inhibited epicardial cell migration (Figure R1F, G).

These *in vivo* data all indicate that $EZH2$ deficiency leads to epicardial migration disorder.

Figure R1

2. Epicardial migration but not proliferation/anti-cell death is claimed to be the major role of $EZH2$. However, the data to support this claim are very weak. Again, no *in vivo* data and only weak *in vitro* data. Careful migration, proliferation and cell death assays are needed.

Response:

Thank you very much for your insightful comments. In this study, terminal deoxynucleotidyl transferase (TdT) dUTP Nick-End Labeling (TUNEL) staining revealed that apoptosis was undetectable in both $EZH2^{epi-KO}$ and $WT1^{GFPCre/+};EZH2^{fl/+}$ hearts, indicating apoptosis is not involved in the decrease of EPDCs in $EZH2^{epi-KO}$ myocardium (Figure R2A).

According to your valuable suggestion, to investigate whether $EZH2$ is involved in epicardial cell proliferation, we performed immunostaining to tested the

co-localization of GFP and proliferating markers (including phosphorylated histone H3, Ki67, and EdU) in EZH2^{epi-KO} mouse epicardium. The immunostaining results showed EZH2 deletion decreased the number of EdU⁺, pH3⁺, Ki67⁺ GFP-positive epicardial cells, indicating epicardial cell proliferation decreased in EZH2^{epi-KO} hearts (Figure R2B). Therefore, we do not rule out the effect of EZH2 specific knockout on the proliferation ability of epicardial cells. However, the novel finding in our paper is that EZH2 deletion regulates epicardial cell migration during embryonic heart development.

Further efforts will be made to elucidate more effects of EZH2 on epicardial cells. At the same time, we explained this situation in the discussion section according to the suggestions of reviewers. This suggestion makes the interpretation of the results of the paper more rigorous. Thank you very much.

Figure R2

3. Timp3 as a possible target of EZH2 is interesting. Yet only Timp3 expression data is present in the manuscript. To make a rather solid conclusion, evidence to show that EZH2 binds to Timp3 promoter and changes histone acetylation/other modifications would be minimally required.

Response:

Thank you very much for your insightful comments. Enhancer of zeste homolog 2 (EZH2) is the catalytic subunit of polycomb repressive complex 2 (PRC2) that

catalyzes tri-methylation of histone H3 at Lys 27 (H3K27me3) and subsequently suppresses transcription of genes bound by such histones [1]. To confirm whether EZH2 directly regulates the expression of Timp3, we conducted the following experiments:

1) We performed ChIP followed by quantitative PCR (ChIP-qPCR) and found that H3K27me3 was enriched at Timp3 promoters in epicardial cell line (Figure R3A, B). The results indicate that the expression of Timp3 may be regulated by PRC2 complex.

2) We performed ChIP-qPCR and found that suppressor of zeste 12 (Suz12, the component of PRC2 complex) was enriched at Timp3 promoters in epicardial cell line (Figure R3C). The results reveal that Timp3 expression was regulated by PRC2 complex. EZH2 is the catalytic subunit of PRC2, whose methyltransferase activity requires two other subunits: embryonic ectoderm development (EED) and Suz12 [2, 3].

3) Together, using two ChIP experiments, we confirmed that Timp3 expression is regulated by PRC2 complex-H3K27me3 modification, in which EZH2 plays a key role. In many literatures, the regulation of EZH2 on genes has been verified by this method [4-7].

Figure R3

[1] Zovoilis Athanasios, *et al.* Destabilization of B2 RNA by EZH2 Activates the Stress Response. *Cell*. 2016.

[2] Cao, R., *et al.* SUZ12 is required for both the histone methyltransferase activity and the silencing function of the EED-EZH2 complex. *Molecular cell*. 2004.

[3] Margueron, R., *et al.* Role of the polycomb protein EED in the propagation of repressive histone marks. *Nature*. 2009.

[4] Yue, Z., *et al.* PDGFR- β Signaling Regulates Cardiomyocyte Proliferation and Myocardial Regeneration. *Cell reports*. 2019.

[5] Liu C., *et al.* PRC2 regulates RNA polymerase III transcribed non-translated RNA gene transcription through EZH2 and SUZ12 interaction with TFIIC complex. *Nucleic Acids Res.* 2015.

[6] Li Z.W., *et al.* PRMT1-mediated EZH2 methylation promotes breast cancer cell proliferation and tumorigenesis. *Cell Death Dis.* 2021.

[7] Sang Ah Yi, *et al.* Transcriptomics-Based Repositioning of Natural Compound, Eudesmin, as a PRC2 Modulator. *Molecules*. 2021.

4. The claim that EZH2 is involved in human epicardial development also lacks real data support. The presence of its expression in human epicardial cells is only a piece of weak association evidence.

Response:

Thank you very much for noting this point of confusion. We performed immunohistochemistry to show EZH2 and H3K27me3 are detectable in human epicardial cells during heart development, which lacks strong data to claim that EZH2 involves in human epicardial development. According to your nice suggestion, we have modified this expression in the revised manuscript from “We found that WT1 (Wilms tumor 1, marker of epicardial cell), EZH2 and H3K27me3 are detectable in human epicardial cells during heart development (Figure 1A), indicating EZH2 mediated H3K27 trimethylation might be essential in human epicardium development.” to “We found that WT1 (Wilms tumor 1, marker of epicardial cell), EZH2 and H3K27me3 are detectable in human epicardial cells during heart development (Figure 1A)”. This suggestion makes the interpretation of the results of the paper more rigorous. Thank you very much.

Reviewer #2 (Comments to the Authors (Required)):

In this study, the investigators have applied immunostaining, qRT-PCR, scratch assay and loss-of -function studies on epicardial cells and found EZH2 is required for mice embryonic survival and epicardial cell migration. The authors then further determined EZH2 influenced epicardial cell migration through regulating ECM components by RNA-seq, ECM mass spectrometry and ChIP-qPCR. Overall, this is a clearly written manuscript with experimental results supporting most of the findings. The reviewer only has several minor concerns.

Figure 2B: Please consider to show a survival curve of the EZH2epi-ko embryos only instead of the bar chart of each genotype, since the figure legend shows "survival of EZH2epi-ko embryos". The ratio of other genotypes can be included in the supplementary data.

Response:

We are very grateful for your comments on the manuscript. According to your nice suggestion, we have supplemented the survival ratio of all genotypes in the data and revised the figure legend (as shown in Figure R4 below).

Figure R4

Figure 2C: Did the authors examine EZH2 expression in EZH2^{epi}-ko embryos at earlier stages, like E10.5? What is the knockout efficiency would be? At what stage does WT1GFP^{Cre} become active?

Response:

Thank you very much for your insightful comments. Based on our data of pattern of WT1, EZH2 and H3K27me3 expression during mouse epicardium development from E10.5 to E13.5, WT1 expression become more active since E11.5-E12.5 (Figure R5A). Therefore, we selected the time point after WT1 activation to detect knockout efficiency in WT1^{GFP^{Cre}+/+};EZH2^{fl/fl} (EZH2^{epi}-KO) mice. By E12.0-E12.5, EZH2^{epi}-KO showed that epicardial cells exhibited markedly reduced EZH2 immunoreactivity, with knockout efficiency of more than 99% (Figure R5B, C).

Figure R5

Figure 3G: For the RNA-seq, is the 1ug of total RNA obtained from a single heart or a pool of several hearts?

Response:

Thank you very much for your comments. The sample of primary epicardial cells for the RNA-seq is isolated from a pool of several (5-6 hearts) E11.5 embryos hearts.

Figure 4B: Is there any difference in Timp3 protein expression level between WT and EZH2epi-ko embryos at E12.5? It can be done by immunostaining of Timp3 antibody.

Response:

Thank you very much for your insightful comments. According to your nice suggestion, we have supplemented the following experiments:

1) We use RNAscope to conduct in situ detection of TIMP3, this result showed that TIMP3 is upregulated in E12.5 EZH2^{epi-KO} epicardial cells compared to littermate controls (Figure R6A).

2) The epicardial cells were cultured with a siRNA transfection kit for another 3 days to knock down EZH2. The cells were collected later to detect the level of TIMP3 protein. We found that EZH2 deletion could cause the increase of TIMP3 protein level (Figure R6B). The hearts dissected from E12.5 EZH2^{epi-KO} embryos were collected but the amount of tissue was too small to reach the level of western blot detection.

Our results indicated that EZH2 deletion could cause the increase of TIMP3 expression level. We have added these results in the revised manuscript. This experiment makes the paper more rigorous and greatly improves the quality of the paper. Thank you again.

Figure R6

Figure 4L/M: How many cells were used for each ChIP? Panel M, is the average number of Actb (negative control) less than 1? Why the sample size for ChIP-qPCR is not consistent?

Response:

Thank you very much for your insightful comments. For each ChIP, cells were harvested from 25 mm live-cell culture dishes at ~75–85% confluence (approximately 1×10^7 cells). As for the average number of Actb and inconsistent sample size, thank you very much for noting this point of confusion, we have checked our original data and found that a clerical error leads to this mistake. We have corrected this result in the revised manuscript (as shown in Figure R7 below). This suggestion makes the interpretation of the results of the paper more rigorous. Thank you very much.

Figure R7

Reviewer #3 (Comments to the Authors (Required)):

In this manuscript, the authors investigated the role of Ezh2 in the developing epicardium They use Wt1 lineage-specific deletion of Ezh2 to reveal an essential role

during embryonic heart development and specifically suggest that epicardial cell migration is compromised. They reveal that Timp3 is upregulated in Ezh2KO epicardial cells, whereas enhanced Timp3 prohibits extracellular matrix degradation and compromises epicardial cell migration. This work is of interest and extends the understanding of Ezh2 in heart development, specifically acting on the epicardium. However, there are some concerns with the study in its current form that ought to be addressed:

In Figure 1, the authors detected Ezh2, H3K27me3 and WT1 expression in human and mouse embryonic epicardium with immunostaining. However, the authors did not show a correlation between Ezh2 and H3K27me3 in epicardial cells. Wt1 is the only marker used to label epicardium. To prove the expression of Ezh2 and H3K27me3 expression/loss in epicardial cells further epicardial markers should be used.

Response:

We are very grateful for your comments on the manuscript. Your suggestions are thoughtful and helpful. In Figure 1, we performed immunohistochemistry to show EZH2 and H3K27me3 are detectable in human and mouse epicardial cells during heart development. EZH2 is a specific histone methyltransferase of histone H3 at Lys 27 (H3K27), EZH2 catalyzes the trimethylation of histone 3 lysine 27 (H3K27me3) [1, 2], leading to the repression of gene transcription and regulating proliferation and differentiation in early embryonic development [3, 4]. The role of the H3K27 methyltransferase EZH2 in the methylation of H3K27 is well established, causing an increase in H3K27me3. According to your suggestion, we have added this explanation of the relationship between EZH2 and H3K27me3 to the results. Your suggestions have increased the readability of our article and made it easier for readers to understand the relationship between EZH2 and H3K27me3.

According to your suggestion, for the problem of label epicardium, we have chosen another epicardial marker, RALDH2, was selected for immunostaining, the results showed that RALDH2 can label epicardial cells (Figure R8A) and the result was consistent with that of WT1 (Figure R8B). Several cell markers used thus far to trace epicardial cells have been derived from genes Wt1, Raldh2, Tbx18, Scleraxia (Scx), and Semaphorin 3D (Sema3D) [5]. WT1 has been showed to be a reliable cell marker for epicardial cells and well investigated in previous literature [5-9]. Based on our findings and the literatures, we selected WT1 as an epicardial cell marker. Thank you again for your positive comments and valuable suggestions to improve the quality of our manuscript.

Figure R8

- [1] Zovoilis Athanasios, *et al.* Destabilization of B2 RNA by EZH2 Activates the Stress Response. *Cell*. 2016.
- [2] R. Margueron, D. Reinberg, The Polycomb complex PRC2 and its mark in life, *Nature*. 2011.
- [3] X. Yin, *et al.* The role and prospect of JMJD3 in stem cells and cancer. *Biomed. Pharmacother.* 2019.
- [4] R. Duan, *et al.* EZH2: a novel target for cancer treatment, *J. Hematol. Oncol.* 2020.
- [5] Cao, Y., Duca, S. & Cao, J. Epicardium in Heart Development. *Cold Spring Harbor perspectives in biology*. 2019.
- [6] Moore, A.W. *et al.* YAC transgenic analysis reveals Wilms' tumour 1 gene activity in the proliferating coelomic epithelium, developing diaphragm and limb. *Mech. Dev.* 1998.
- [7] Zhou, B. *et al.* Epicardial progenitors contribute to the cardiomyocyte lineage in the developing heart. *Nature*. 2008.
- [8] Zhou, B. & Pu, W. T. Genetic Cre-loxP assessment of epicardial cell fate using Wt1-driven Cre alleles. *Circulation research*. 2012.
- [9] Velecela, V. *et al.* Epicardial cell shape and maturation are regulated by Wt1 via transcriptional control of Bmp4. *Development (Cambridge, England)*. 2019.

2) Wt1 is expressed in the intermediate mesenchyme from E9.0 and is also expressed in the coronary endothelium (Armstrong JF *et al.*, 1992). Thus, the phenotype of *Ezh2* deletion in *Wt1-Cre* is not only due to *Ezh2* loss in the epicardial lineage but also other cell types, such as endothelial cells. For example, the defects shown in Fig.2D, H, L and K could be the direct consequence of *Ezh2* loss in CD31+ cells rather than *Ezh2* loss in epicardial cells. To exclude this possibility, the authors should use an

inducible Cre, such as Wt1-CreErt2 (Zhou B et al., 2008; Zhou B and Pu W, 2012) to delete Ezh2 in the epicardium more specifically.

Response:

Thank you very much for your insightful comments. According to your valuable suggestion, we have supplemented the following experiments:

1) We constructed inducible EZH2 epicardial conditional knockout (cKO) mouse by crossing WT1^{CreERT2/+} mouse line with EZH2^{f/f}. Two doses of Tamoxifen were administered at E9.5 and E10.5, WT1^{CreERT2/+}; EZH2^{f/f} (hereafter as Ezh2^{epi-iKO}) epicardial cells exhibited markedly reduced EZH2 immunoreactivity by E12.5 (Figure R9A, B). EZH2^{epi-iKO} mouse could survive to E15.5. Immunostaining results showed a less density of intramyocardial vessels, decreased epicardial cell number, and inhibited epicardial cell migration (Figure R9C-G.), with more deposit of laminin and collagen type IV at the basement membrane (Figure R9H, I). This cardiac phenotype of EZH2^{epi-iKO} mice is generally consistent with that of EZH2^{epi-KO} mice.

2) We analyzed the endothelial cell-specific EZH2 KO efficiency by co-immunostaining of EZH2/H3K27me3 with CDH5 in EZH2^{epi-KO} mice. There was a large amount of EZH2⁺ or H3K27me3⁺ endothelial cell in WT1^{GFP^{Cre}/+};EZH2^{f/+} and WT1^{GFP^{Cre}/+};Ezh2^{f/f} hearts, and there was no significant difference between the two groups (Figure R9J, K). The results suggested that epicardial cell specific EZH2 knockout had no effect on EZH2 expression in endothelial cells.

We have added these results in the supplementary material. This experiment makes the paper more rigorous and greatly improves the quality of the paper. Thank you again.

Figure R9

3) Aside from the reported migration defects, the authors have not characterized in sufficient depth the epicardial defects of *Ezh2*KO mutants. For example, the authors should evaluate epicardial cell proliferation and quantify WT1⁺ cells in the epicardium and WT1⁺ EPDCs in the myocardium; alongside further epicardial markers such as podoplanin or Integrin alpha4/beta1.

Response:

Thank you very much for your insightful comments. According to your suggestion, we evaluate epicardial cell proliferation and quantify WT1⁺ cells in the epicardium.

1) We found that the WT1-positive epicardial cell count was significantly decreased in E12.5 EZH2^{epi-KO} hearts (Figure R10A, B).

2) In our experiment, GFP reporter in WT1^{GFP^{Cre/+};EZH2^{f/f}} was also an ideal tool for locating epicardial cells, labeled epicardium in genetically. To investigate whether EZH2 is involved in epicardial cell proliferation, we performed immunostaining to tested the co-localization of GFP and proliferating markers (including phosphorylated histone H3, Ki67, and EdU) in EZH2^{epi-KO} mouse epicardium. Our results showed that a reduced proliferation in EZH2^{epi-KO} hearts compared to littermate controls at E10.0 (Figure R10C-F).

The above results suggest that EZH2 regulates the proliferation of epicardial cells is also worth studying in the future, and further efforts will be made to elucidate more effects of EZH2 on epicardial cells proliferation and migration. At the same time, we explained this situation in the discussion section according to the suggestions of reviewers. Thank you again for your positive comments and valuable suggestions to improve the quality of our manuscript.

Figure R10

4) The authors detected Timp3 upregulation based on RNA profiling, however, they should also determine whether Timp3 is upregulated in Ezh2 knockout epicardium using an in situ RNA-based approach (spatial HCR, RNAscope) and at the level of the protein via immunostaining.

Response:

Thank you very much for your insightful comments. According to your nice suggestion, we have supplemented the following experiments:

1) We use RNAscope to conduct in situ detection of TIMP3, this result showed that TIMP3 is upregulated in E12.5 EZH2^{epi-KO} epicardial cells compared to littermate controls (Figure R6A).

2) The epicardial cells were cultured with a siRNA transfection kit for another 3 days to knock down EZH2. The cells were collected later to detect the level of TIMP3 protein. We found that EZH2 deletion could cause the increase of TIMP3 protein level (Figure R6B). The hearts dissected from E12.5 EZH2^{epi-KO} embryos were collected but the amount of tissue was too small to reach the level of western blot detection.

Our results indicated that EZH2 deletion could cause the increase of TIMP3 expression level. We have added these results in the revised manuscript. This experiment makes the paper more rigorous and greatly improves the quality of the paper. Thank you again.

Figure R6

5) The authors suggest that the absence of Ezh2 results in decrease of H3K27me3 at the promoter of Timp3, however, Figures 3 and 4 do not address whether this is cause

or effect on the altered Timp3 expression in the KOs, which needs addressing. The authors should also include whether Timp3 is detected in the MS analysis and GSEA (Figure 4N and 4O).

Response:

Thank you very much for your insightful comments. According to your nice suggestion, we have supplemented the following experiments. The epicardial cells were cultured with a siRNA transfection kit for 72h to knock down TIMP3. The expression of EZH2 and TIMP3 was detected by western blot. The results showed that the expression of EZH2 was not affected by TIMP3 knockdown (Figure R11A). Meanwhile, the epicardial cells knocked down EZH2 with siRNA resulted in increased expression of TIMP3 (Figure R11B), suggesting that the absence of EZH2 cause the altered TIMP3 expression.

Figure R11

Reviewer #4 (Comments to the Authors (Required)):

In this elegantly designed study by Jiang et al demonstrates a crucial role of EZH2 in epicardial cell migration and cardiac development. The data are high quality and properly interpreted, and support the main conclusion in general. Overall, this is a nice manuscript with interesting findings; however, a few questions should be clarified.

Specific comments:

1. In all figures, the embryos of $WT1^{GFPcre/+};EZH2^{f/+}$ was used as controls which exhibited normal epicardial cell migration and heart development, suggesting 50% of $EZH2$ is sufficient to maintain the proper function of epicardial cells. In contrast, 50% reduction of $EZH2$ in cultured epicardial cells showed dramatic phenotype. This discrepancy should be explained.

Response:

Thank you very much for noting this point of confusion. In our study, the expression pattern of $EZH2$ in epicardium of heterozygous mice was nearly the same as that in wild-type. Previous findings have showed that mice that are heterozygous for $EZH2$ are phenotypically normal, suggesting compensatory mechanisms working to maintain expression of $EZH2$ in these cells. [1-3]. Consistent with previous studies, we observed that heterozygous $WT1^{GFPcre/+};EZH2^{f/+}$ mice did not differ in the physiological processes of growth and development to reproduction and defense from $EZH2^{f/f}$ mice and wild-type mice. Further, we detected the mRNA level of $EZH2$ by qRT-PCR in epicardial cells isolated from $EZH2^{flox/flox}$ mice and $WT1^{GFPcre/+};EZH2^{f/+}$ mice, which showed that there were no significant gene expression differences observed between the groups (Figure R12). Heterozygous controls were used in this study based on prior findings of only negligible differences between heterozygous and wild-type subjects.

Figure R12

[1] Delgado-Olguín, P. et al. Epigenetic repression of cardiac progenitor gene expression by *Ezh2* is required for postnatal cardiac homeostasis. *Nat. Genet.* 2012.

[2] He, A. et al. Polycomb repressive complex 2 regulates normal development of the mouse heart. *Circulation research.* 2012.

[3] Chen, L. et al. Conditional ablation of *Ezh2* in murine hearts reveals its essential roles in endocardial cushion formation, cardiomyocyte proliferation and survival. *PLoS one.* 2012.

2. Since EZH2 is essential for cell proliferation of cardiomyocyte as introduced at begin, it will be interesting to check whether reduced cell proliferation contributed to the cardiac phenotype?

Response:

Thank you very much for your insightful comments. According to your valuable suggestion, to investigate whether EZH2 is involved in epicardial cell proliferation, we performed immunostaining to tested the co-localization of GFP and proliferating markers (including phosphorylated histone H3, Ki67, and EdU) in EZH2^{epi-KO} mouse epicardium. The immunostaining results showed EZH2 deletion decreased the number of EdU⁺, pH3⁺, Ki67⁺ GFP-positive epicardial cells, indicating epicardial cell proliferation decreased in EZH2^{epi-KO} hearts (Figure R2B). Therefore, we do not rule out the effect of EZH2 specific knockout on the proliferation ability of epicardial cells. However, the novel finding in our paper is that EZH2 deletion regulates epicardial cell migration during embryonic heart development.

The above results suggest that EZH2 regulates the proliferation of epicardial cells is also worth studying in the future, and further efforts will be made to elucidate more effects of EZH2 on epicardial cells proliferation and migration. At the same time, we explained this situation in the discussion section according to the suggestions of reviewers. Thank you again for your positive comments and valuable suggestions to improve the quality of our manuscript.

Figure R13

3. Mechanistically, the authors showed that knockdown of EZH2 in cultured epicardial cells upregulated the expression of TIMP3. Consistently, Laminin and Collagen IV, two targets of TIMP3, were increased in the EZH2 KO hearts. The authors concluded that EZH2-dependent suppression of TIMP3 is essential for the degradation of ECM and epicardial cell migration. Since TIMP3 is the key downstream mediator of EZH2, the in vivo data is needed to show the upregulation of TIMP3 in EZH2 knockout hearts.

Response:

Thank you very much for your insightful comments. According to your nice suggestion, we have supplemented the following experiments:

1) We use RNAscope to conduct in situ detection of TIMP3, this result showed that TIMP3 is upregulated in E12.5 EZH2^{epi-KO} epicardial cells compared to littermate controls (Figure R6A).

2) The epicardial cells were cultured with a siRNA transfection kit for another 3 days to knock down EZH2. The cells were collected later to detect the level of TIMP3 protein. We found that EZH2 deletion could cause the increase of TIMP3 protein level (Figure R6B). The hearts dissected from E12.5 EZH2^{epi-KO} embryos were collected but the amount of tissue was too small to reach the level of western blot detection.

Our results indicated that EZH2 deletion could cause the increase of TIMP3 expression level. We have added these results in the revised manuscript. This experiment makes the paper more rigorous and greatly improves the quality of the paper. Thank you again.

Figure R6

4. The deletion of EZH2 in epicardium severely impeded the myocardial growth, suggesting EZH2-dependent paracrine signals in epicardium is essential for myocardial development. However, this was not studied or discussed in the paper.

Response:

Thank you very much for your insightful comments. A previous study found that mouse embryos with epicardial malformations caused by WT1 mutations would have ventricular dysplasia, especially with the thinning of the dense ventricular layer. This suggests a relationship between the adjacent epicardium and the myocardium, whose proliferation and development may be regulated by the epicardium and paracrine effect [1]. Further evidence includes: Pennisi DJ et al. found that the proliferation activity of the myocardial layer is decreasing from the outer layer to the inner layer, and the surgical peeling of the epicardium of chicken embryos will lead to abnormal

thinning of the myocardial layer, which also suggests that the epicardium may secrete some auxin to regulate the proliferation of the myocardial layer, so the myocardial proliferation ability changes with the diffusion distance [2].

In our study, we found that EZH2 epicardial deletion impeded epicardial cell migration, which resulted in myocardial hypoplasia and defective coronary plexus development. To explore the mechanism of EZH2 in regulating epicardial cells, we analyzed the gene expression profile by RNA-Seq. Gene Ontology (GO) analysis showed that genes were enriched in terms of cell migration (Figure R14), which showed no signs of paracrine signal regulation in EZH2-deficient epicardial cells. Based on these data, we supposed that dysregulated paracrine signals in epicardium for myocardial development might be secondary to loss of EPDCs in myocardium. At the same time, we explained this situation in the discussion section according to the suggestions of reviewers. This suggestion makes the interpretation of the results of the paper more rigorous. Thank you very much.

Enriched Gene Ontology Terms	
Term	BH p-val
system development	1.5E-07
regulation of cell migration ←	4.2E-06
regulation of locomotion	1.8E-06
regulation of developmental process	7.6E-05
epithelium development	6.8E-04
tissue development	6.8E-04
negative regulation of cell migration ←	3.8E-03
regulation of epithelial cell migration ←	2.4E-02

Figure R14

[1] Duim SN, Goumans MJ and Kruithof BPT. WT1 in Cardiac Development and Disease. In: M. M. van den Heuvel-Eibrink, ed. Wilms Tumor Brisbane (AU): Codon Publications.

[2] Pennisi DJ, Ballard VL and Mikawa T. Epicardium is required for the full rate of myocyte proliferation and levels of expression of myocyte mitogenic factors FGF2 and its receptor, FGFR-1, but not for transmural myocardial patterning in the embryonic chick heart. Developmental dynamics: an official publication of the American Association of Anatomists. 2003.

5. The images in Figure4P, Q showed that the expression of Laminin and Collagen IV was increased in both sub-epicardium and sub-endocardium of mutant embryos,

whereas the deletion is epicardial-specific, suggesting that the upregulation of those ECM proteins may not be specifically due to the loss of EZH2.

Response:

Thank you very much for your insightful comments. As TIMP3 is a secretory protein, which diffuses into the surrounding extracellular matrix and regulates degradation of the extracellular matrix. As a consequence, apart from the obvious differences of expression of Laminin and Collagen IV in sub-epicardium, similar modest differences in sub-endocardium and myocardium can also be found.

Minor point: The font size in Figure 4O is too small.

Response:

We are very grateful for your comments on the manuscript. We have modified it according to your suggestion. Due to the length of the article, we adjust Figure 4O as the supplementary materials (Figure S3E) in the revised manuscript.

March 15, 2023

RE: Life Science Alliance Manuscript #LSA-2022-01765-TR

Shengshou Hu

State Key Laboratory of Cardiovascular Disease, Fuwai Hospital, National Center for Cardiovascular Disease, Chinese Academy of Medical Sciences and Peking Union Medical College

Dear Dr. Hu,

Thank you for submitting your revised manuscript entitled "EZH2 controls epicardial cell migration during heart development". We would be happy to publish your paper in Life Science Alliance pending final revisions necessary to meet our formatting guidelines.

- please consider the final Reviewer's 1 suggestion
- please add a separate Data Availability section in which you should add your RNA seq data (repository and accession number)
- please upload both your main and supplementary figures as single files
- please upload your table files as editable doc or excel files or make sure that the tables are in your doc file of your main manuscript text
- please upload your graphical abstract as a separate file, labeled as graphical abstract
- please add ORCID ID for first corresponding author-you should have received instructions on how to do so

Figure Check:

- please increase the visibility of your scale bars throughout

A. FINAL FILES:

B. MANUSCRIPT ORGANIZATION AND FORMATTING:

Sincerely,

Reviewer #1 (Comments to the Authors (Required)):

The authors did a very nice job of addressing all the reviewer critiques and the revised manuscript has been significantly improved. Overall, the conclusion that EZH2 is essential for epicardial cell migration into myocardium is convincing. The mechanistic explanation that EZH2 suppresses Timp3 transcription to controls the integrity of basement membrane and therefore migration of epicardial cells is reasonable.

One major issue is that there are very minor changes in the original figures with the additional data presented mostly as supplementary figures. The suggestion is to carefully integrate most of the supplementary data into major figures.

Reviewer #2 (Comments to the Authors (Required)):

The authors have answered all my questions, the reviewer has no further concerns.

Reviewer #3 (Comments to the Authors (Required)):

In this study, Jiang et al investigate the role of EZH2 in epicardial cell migration. Using a constitutive and an inducible Wt1-Cre line, they delete Ezh2 in epicardium and prove it to be essential for heart development and epicardial cell proliferation and migration. They identify TIMP3 as a target of EZH, confirm the binding of SUZ12, another component of the PRC2 complex, on TIMP3's promoter, and confirm enrichment of repressive histone code. Loss of EZH2 results in TIMP3 up-regulation. They also suggest that enhanced TIMP3 expression inhibits ECM degradation and epicardial cell migration. This study extends the understanding of EZH2 in embryonic heart development and brings new insights on epicardial cell migration and ECM dynamics at the epigenetic level.

Reviewer #4 (Comments to the Authors (Required)):

The authors have done great work and addressed all my concerns.

March 27, 2023

RE: Life Science Alliance Manuscript #LSA-2022-01765-TRR

Dr. Yu Nie

State Key Laboratory of Cardiovascular Disease, Fuwai Hospital, National Center for Cardiovascular Disease
State Key Laboratory of Cardiovascular Disease, Fuwai Hospital, National Center for Cardiovascular Disease, Chinese
Academy of Medical Sciences and Peking Union Medical College
State Key Laboratory of Cardiovascular Disease, Fuwai Hospital, National Center for Cardiovascular Disease, Chinese
Academy of Medical Sciences and Peking Union Medical College
State Key Laboratory of Cardiovascular Disease, Fuwai Hospital, National Center for Cardiovascular Disease, Chinese
Academy of Medical Sciences and Peking Union Medical College
Beijing 100037
China

Dear Dr. Nie,

Thank you for submitting your Research Article entitled "EZH2 controls epicardial cell migration during heart development". It is a pleasure to let you know that your manuscript is now accepted for publication in Life Science Alliance. Congratulations on this interesting work.

DISTRIBUTION OF MATERIALS:

Again, congratulations on a very nice paper. I hope you found the review process to be constructive and are pleased with how the manuscript was handled editorially. We look forward to future exciting submissions from your lab.

Sincerely,
